# DRIVE v1.0: A data-driven framework to estimate road transport emissions and temporal profiles

Daniel Kühbacher[1], Jia Chen[1], Patrick Aigner[1], Mario Ilic[3], Ingrid Super[2], and Hugo Denier van der Gon[2]

[1]Professorship of Environmental Sensing and Modeling, Technical University of Munich
[2]Department of Climate, Air and Sustainability, TNO, Utrecht, Netherlands
[3]Chair of Traffic Engineering and Control, Technical University of Munich

**Correspondence:** Daniel Kühbacher (daniel.kuehbacher@tum.de) and Jia Chen (jia.chen@tum.de)

**Abstract.** Traffic in urban areas is an important source of greenhouse gas (GHG) and air pollutant emissions. Estimating traffic-related emissions is therefore a key component in compiling a city emission inventory. Inventories are fundamental for understanding, monitoring, managing, and mitigating local pollutant emissions.

We present DRIVE v1.0, a data-driven framework to calculate road transport emissions based on a multi-modal macroscopic traffic model, vehicle class-specific traffic counting data from more than a hundred counting stations, and HBEFA emission factors. DRIVE introduces a novel approach for estimating traffic emissions with vehicle-specific temporal profiles in hourly resolution. In addition, we use traffic counting data to estimate the uncertainty of traffic activity and the resulting emission estimates at different temporal aggregation levels and with road link resolution. The framework was applied to the City of Munich, covering an area of 311 $km^2$ and accounting for GHGs ($CO_2$, $CH_4$) and air pollutants ($PM$, $CO$, $NO_x$). It captures irregular events such as COVID lockdowns and holiday periods well and is suitable for use in near real-time applications. Emission estimates for 2019-2022 are presented and differences in city totals and spatial distribution compared to the official municipal reported and national and European downscaled inventories are examined.

## 1 Introduction

Urban areas are centers of human activity that contribute to high greenhouse gas (GHG) emissions and poor air quality. As the global population becomes increasingly urbanized, cities face the dual challenge of supporting economic growth and development while mitigating environmental impacts. 75 % of the world's primary energy, mainly in the form of fossil fuels, is consumed in cities, resulting in 70 % of global $CO_2$ emissions (IEA, 2024). Approximately 20-50 % of urban greenhouse gases and air pollutants are associated with transportation, mainly from road vehicles (Chapman, 2007; Edenhofer et al., 2014; Crippa et al., 2021). These emissions not only exacerbate climate change but also pose a severe health risk to the urban population through exposure to poor air quality. The European Environment Agency (EEA) attributes 307,000 premature deaths to chronic exposure to particulate matter and 40,400 deaths to nitrogen dioxide exposure in Europe (EEA, 2022). Understanding the sources, magnitude, and trends of transport emissions is crucial for developing effective mitigation strategies. A road transport emission inventory provides a detailed assessment of emissions from different transport modes, enabling the identification of

key sources and the evaluation of mitigation measures (Ting Wei et al., 2021; Arioli et al., 2020). In the German Climate Action
Plan, a mid-term goal has been set to reduce GHG emissions from the transport sector by 40-42 % by 2030 compared to 1990
levels (BMUB, 2016). The City of Munich aims to reduce traffic-related GHG emissions by 58 % by 2035 compared to 2018
levels (Timpe et al., 2021). Moreover, the World Health Organization (WHO) has lowered the annual threshold values for $NO_2$
from 40 $\mu g/m^3$ to 10 $\mu g/m^3$ and for $PM_{10}$ from 20 $\mu g/m^3$ to 15 $\mu g/m^3$ in its current air quality guidelines (WHO, 2021).
Meeting these ambitious goals demands accurate emission estimates as an essential resource for policymakers, urban planners,
and environmental scientists.

A sub-kilometer spatial resolution and hourly temporal resolution become essential when inventories are used alongside ob-
servations and atmospheric transport models. Methodologies exist to disaggregate nationally reported emissions on spatial
and temporal scales using proxy information (e.g., population count, road density). Popular examples of inventories following
that top-down approach utilizing different spatial proxy maps are ODIAC (Oda and Maksyutov, 2011; Oda et al., 2018), and
CAMS-REG-v4 (Kuenen et al., 2022). The German Federal Environmental Agency (UBA) also provides gridded emissions for
Germany using national proxy data or spatially disaggregated activity data (Schneider et al., 2016). However, these top-down
emission distribution approaches are highly uncertain on a cell scale (Super et al., 2020) and might instead reflect the emission
proxy more than the actual emissions (Hutchins et al., 2017).

For the transport sector, data from traffic counting stations, traffic models, and driving patterns are used to accurately capture
local activity. Methods based on this granular activity information are generally called bottom-up methods. The activity data
can be combined with a range of emission models such as HBEFA (Handbook Emission Factors for Road Transport) (Notter
et al., 2019, 2022), COPERT (Computer Programme for the Calculation of Emissions from Road Transport) (Ntziachristos and
Samaras, 2024) or PHEM (Passenger Car and Heavy Duty Emission Model) (Hausberger, 2003) to enable bottom-up emission
estimation. They can be classified based on how they incorporate traffic behavior, though more complex emission models do
not necessarily perform systematically better (Smit et al., 2010). HBEFA and COPERT use average speeds or distinct traffic
situations and road types to consider average traffic behavior. PHEM takes a more detailed approach by incorporating second-
by-second vehicle operating data to reflect instantaneous driving conditions.

Various frameworks also implement these emission models to estimate road transport emissions at high spatiotemporal resolu-
tion. Each is designed for distinct target areas, user groups, and applications. VEIN is a particularly comprehensive emission
modeling R-package that implements COPERT, emission factors from Brazil, China, as well as a database interface to MOVES
(USEPA, 2024). It has applications in developing countries and includes the Carter (2015) methodology to group species into
chemical mechanisms (Ibarra-Espinosa et al., 2018). DARTE provides a national on-road $CO_2$ inventory for the US based
on average annual daily traffic counts (AADT) from highway counting stations (Gately et al., 2015). HERMESv3 includes a
bottom-up module with a coupled macroscopic traffic and emission modeling system targeting the Barcelona metropolitan area
(Rodriguez-Rey et al., 2021; Guevara et al., 2021). CARS (Baek et al., 2022) and Yeti (Chan et al., 2023) are both Python-
based and were implemented for Seoul (South Korea) and Berlin (Germany), respectively. The traffic models these frameworks
use typically provide road-specific, time-aggregated information for a representative weekday or peak hours. This information
is either used to distribute aggregated emission estimates or temporally extrapolated with generalized traffic activity profiles.

However, they do not leverage the full potential of local activity time series for temporal extrapolation or uncertainty assessment. A more activity-data-intensive approach is to use floating car data (FCD; GPS information of individual vehicles) or information from intelligent traffic monitoring systems (ITMS) (Jiang et al., 2021; Wen et al., 2020; Wu et al., 2020; Yang et al., 2019; Gately et al., 2017). This data provides exact speed and congestion information with high spatiotemporal resolution. Nevertheless, data protection concerns and substantial costs by commercial data providers widely restrict the use and availability of these information sources. In contrast, conventional traffic counting data (e.g., from induction loop meters) is widely available in most cities, highlighting the potential for more extensive use of these datasets for local, data-driven emissions modeling.

With this work, we present a road transport emission inventory framework (DRIVE v1.0; Data-driven Road-Transport Inventory for Vehicle Emissions Version 1.0), written in Python, that combines a comprehensive macroscopic traffic model, vehicle-specific traffic counting data, and HBEFA emission factors. Our novelty is the extensive use of counting data from more than 100 individual traffic detectors across different road types within our city of interest. First, the local counting data is used to extrapolate the traffic model over time to accurately estimate the daily traffic volume on each road segment. Next, we introduce a novel method for data-based and time-resolved calculation of the vehicle share. The vehicle class-specific temporal scaling of traffic activity enables a precise and data-based prediction of traffic conditions to apply traffic-related emission factors. Finally, we assess model uncertainties based on the counting data. With the possibility of accessing local data in real-time, our method is suitable for near-real-time applications, such as the assessment of air quality or greenhouse gas mitigation measures.

The framework was developed and implemented for the City of Munich as part of the ICOS Cities project. ICOS Cities aims to develop and evaluate standardized greenhouse gas measurement and services in urban environments, with Munich as one pilot city. The inventory represents Munich's first spatially and temporally explicit bottom-up traffic inventory, covering all road-based vehicle categories. It accounts for GHG emissions like carbon dioxide (fossil-fuel: $CO_{2,ff}$, fossil- and bio-fuel: $CO_{2,ff+bf}$), methane ($CH_4$), and co-emitted species such as carbon monoxide ($CO$), nitrous oxides ($NO_x$) and particulate matter ($PM_{10}$). Our data-driven approach effectively captures special events such as the COVID-19 pandemic, vacation periods, and Christmas. We compare our estimates to city-reported figures and national and European datasets available for the Munich region. The resulting emission- and uncertainty estimates and temporal profiles are invaluable for the timely evaluation of mitigation measures and atmospheric modeling of greenhouse gases and air pollutants.

## 2  Methodology

The most relevant traffic activity variables for emission modeling are the traffic flow, vehicle-specific information (type, age, category, size), speed, road network, and fuel used (Pinto et al., 2020). Data availability remains the most significant challenge and sets inherent limits to implementing traffic emission inventories (Arioli et al., 2020). The availability of activity data determines the selection of the emission model and the associated emission factors. Our framework exploits traffic data available in our target city, Munich. Subsequently, HBEFA 4.2 emission factors are applied, as they are also utilized and regularly updated

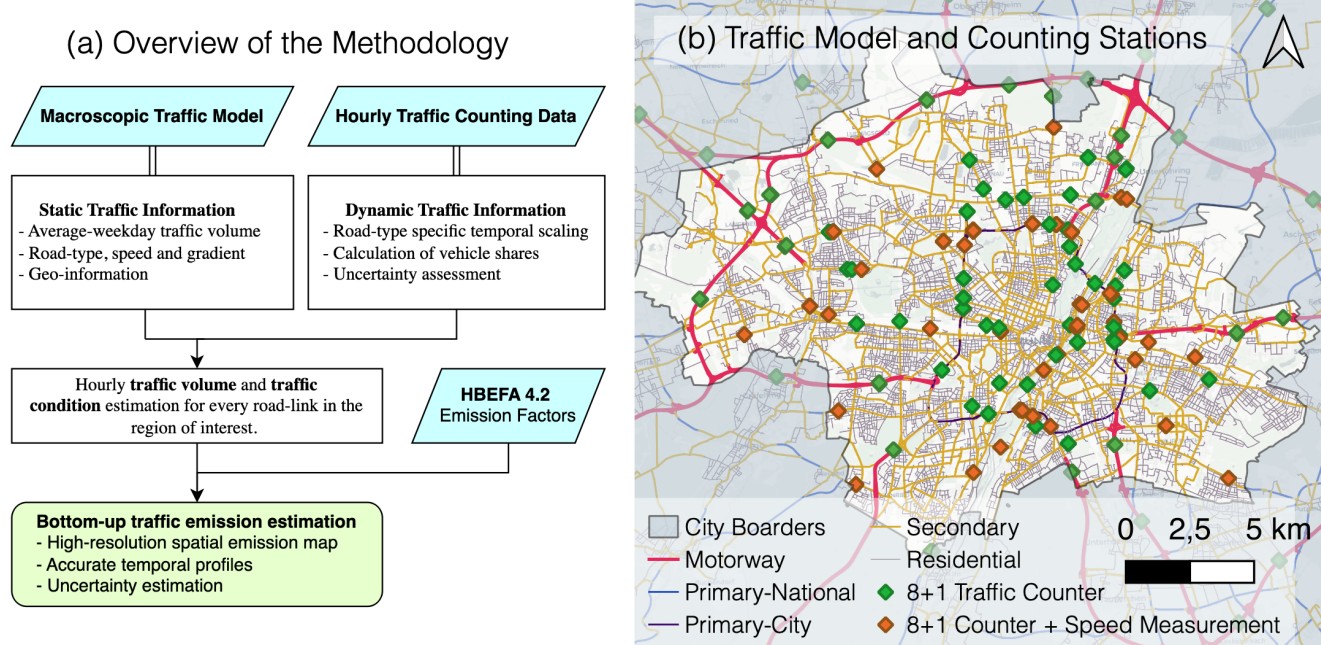

**Figure 1.** Overview of the methodology and data applied in the present study. (a) depicts how static and dynamic traffic activity data is used to calculate the hourly traffic volume and condition. Subsequently, HBEFA emission factors are applied to calculate emissions. (b) provides an overview of the study area and indicates the locations of traffic counting stations in the region of interest. Each traffic counter shown provides vehicle-specific counts, and 43 stations also provide average speed measurements. 116 individual counting stations are available in our area of interest.

by Germany's Federal Environment Agency. These factors incorporate national statistical data on the vehicle fleet, including age, size, fuel type, and EURO emission class parameters. Figure 1a shows an overview of the proposed methodology. Static traffic information derived from a traffic model is combined with dynamic traffic information from traffic counting stations to estimate the hourly traffic volume and condition for every road segment in the model.

## 2.1 Traffic Activity Data

### 2.1.1 Traffic Model

Many medium-sized to large cities maintain macroscopic traffic models for infrastructure planning, simulating traffic volumes, or analyzing travel demand scenarios. They are calibrated with observations like traffic counts, speed measurements, and data from mobility behavior surveys. The City of Munich maintains a macroscopic traffic demand model for the city area and surrounding municipalities based on PTV's (Planung Transport Verkehr GmbH) software VISUM. The model is not publicly available, but was provided free of charge by the city administration after signing a data transfer agreement. The road network is split into road segments, which are represented as lines in the model. The spatial resolution ranges from several tens of

meters in densely networked inner-city areas to a kilometer scale on highways. The year of analysis is 2019. It represents the average annual norm-weekday (Tuesday-Thursday outside the holiday season) traffic volume $q_{i,vc}^{model}$ for passenger cars (PC), light commercial vehicles (LCV), heavy goods vehicles (HGV), and public transport. The model data was exported for our region of interest, providing the traffic volume $q_{i,vc}^{model}$ [veh/day] of the aforementioned vehicle classes, the speed limit $v_i$ [km/h], road type $r_i$, road gradient $s_i$ [%], hourly road capacity $C_i$ [veh/h], and geoinformation for each road segment $i$ of the model.

### 2.1.2 Traffic Counting Data

Camera-based and induction loop traffic counters are frequently used for monitoring and management. We utilize hourly traffic counting data to scale the average weekday traffic volumes from the traffic model to each day within the study period. In addition, the data is used to create diurnal traffic patterns that assign daily traffic volumes to specific hours of the day. Details on the temporal scaling are presented in section 2.2.

In total, 116 traffic counters are available in our region of interest. The city administration maintains 82 stations within the urban region, while the BASt (Bundesagentur für Straßenbau, eng.: Federal Road Agency) provides counting data from 34 locations on national motorways and federal primary roads. Like the traffic model, the traffic counting data from the city administration is not publicly available but was provided free of charge after signing a data transfer agreement. Counting data from the BASt is shared publicly under the CC BY 4.0 license (BASt, n.d.). Both data sources categorize the counts into eight vehicle classes (passenger cars, passenger cars with trailers, light commercial vehicles, heavy trucks, heavy trucks with trailers, semi-trailer trucks, motorcycles, and buses) and one category for unclassified vehicles. Some stations also provide the average speed of vehicles, but this data is not used in the model as it was deemed unreliable. We observed numerous artifacts and outliers in the speed data, which we attributed to stop-and-go traffic, intersection effects, and maintenance issues. Moreover, there is no speed information available for the motorway (BASt counters), which makes it impractical to use this data consistently for all major road types. In figure 1b, the locations of the counting stations are superimposed on Munich's road network. Some stations were excluded because they lacked data for the study period or were densely clustered near specific locations, such as the trade fair area or Allianz Arena (soccer stadium).

### 2.1.3 Data Preprocessing

Manual data curation and data preprocessing are required to make the traffic counting data compatible with the traffic model and to pass on the counting data to the modules provided by the framework. The exact steps required depend on the format and quality of the available data and may differ in other cities. The basic procedure is described in the following section.

The first step is to convert traffic count information from different sources to a standard data model (see Appendix B1). For this, the location of the traffic detectors must also be allocated to the respective road link in the traffic model on which the data is observed. It enables the model data to be combined with counting data and the counting data to be supplemented with road link details. If multiple detectors are located on the same road link, the counting data has to be aggregated subsequently. Additionally, the 8+1 vehicle classification from the traffic counting data needs to be aggregated into vehicle classes that are

compatible with the HBEFA classification as presented in table 1.

**Table 1.** Aggregation of vehicle categories from 8+1 counting data categorization to HBEFA compatible vehicle classes.

| 8+1 vehicle class | HBEFA vehicle class ($vc$) |
|---|---|
| Passenger Car | PC - Passenger Car |
| Passenger Car w. Trailer | |
| Motorcycles | MOT - Motorcycles |
| Light Truck | LCV - Light Commercial Vehicle |
| Truck | HGV - Heavy Goods Vehicles |
| Truck w. Trailer | |
| Truck w. Semi-Trailer | |
| Bus | BUS - Coach |
| Not Classified | - |

A modified z-score (Iglewicz Boris and David C. Hoaglin, 1993) is applied to the daily count value for outlier detection.
The counting data is grouped by vehicle class, day, and road type, and a z-score threshold of $\pm 3.5$ is applied. Additionally, we compare the total daily count and the sum of all hourly counts. If these values do not match, the error is usually due to communication problems between the detectors and the traffic management center (Lu et al., 2008). We defined an error margin of $\pm 5\%$ for the sum of the hourly counts compared to the daily count, outside of which the data rows are removed.

Two additional attributes, *"complete"* and *"valid"*, were added to the counting data to filter out time series that are not suitable
for the temporal extrapolation of the traffic model (see section 2.2). *"Complete"* time series were empirically defined as series that cover more than 80 % of all days in the time period of interest. The *"valid"* property indicates how well the counted traffic volume matches the traffic volume provided by the traffic model. A bad fit is related to improper model calibration, hardware faults of the traffic detectors, wrong assignment of the counting station to the traffic model, or inconsistent detector identifiers. The Scalable Quality Value (SQV) (Friedrich et al., 2019) is used to assess the fit between counting data and the traffic model.
It is a measure between 0 (no fit) and 1 (perfect fit). From the counting data, we calculate the annual average weekday traffic volume for 2019 $q_{i,vc}^{count,ref}$ for each road link with an assigned counting station. This value is representative of the one in the traffic model, allowing for direct comparison. An SQV-threshold of 0.6 covers > 90 % of the stations, which were subsequently flagged as *"valid"*.

## 2.2   Temporal Scaling

This section describes our methodology for extrapolating traffic model information using the preprocessed counting data. First, three different road categories $r_i$ are defined for scaling: Motorway, primary city, and distributor/secondary roads. Residential, local, and distributor roads (i.e., minor roads) are scaled using the same factor because significant differences in activity were

not observed, and there are hardly any detectors on minor roads and residential roads. Next, we define three different day types $d$: Weekday (Monday to Friday), Saturday, and Sunday/Holiday. They all show individual activity behavior. Finally, we incorporate the HBEFA vehicle class categorization (Table 1).

Equation 1 shows the calculation of the hourly traffic volume and is applied to each road link in the traffic model. The $q_{i,vc}^{model}$ is scaled to any day in the time period of interest using the annual cycle of the present road type $\alpha_r$ (details in subsection 2.2.1). We assume that the activity on roads of the same type scales identically. The total traffic volume is then split among all vehicle classes using the road-type dependent vehicle share $\delta_{vc,r}$ (details in subsection 2.2.2). Additional correction factors $\kappa_{i,vc}$ were computed to incorporate spatial differences in vehicle shares (e.g., arterial motorways have a lower HGV share than the ring motorway). Vehicle and day-type individual diurnal cycles $\beta_{vc,d}$ (details in subsection 2.2.3) are applied to obtain an hourly traffic volume $q_{i,vc}$.

$$q_{i,vc} = q_{i,vc}^{model} \cdot \alpha_r \cdot \beta_{vc,d} \cdot \delta_{vc,r} \cdot \kappa_{i,vc} \tag{1}$$

### 2.2.1 Annual Cycles

For scaling, the counting time series are normalized to the reference time period that the traffic model represents. In our implementation, the model represents the average norm weekday of 2019. Therefore, each counting time series was normalized by the average norm-weekday count of 2019 $q_{i,vc}^{count,ref}$. The time series were then grouped by road type, and the median of the normalized daily traffic volume of all *"valid"* and *"complete"* time series was calculated. It results in three separate annual cycles $\alpha_r$ for the road type "Motorway-National", "Trunk Road/ Primary-City", and "Distributor/Secondary", which also covers lower-level road types. Missing values were imputed using an average scaling factor from normalized counts of the identical day type in the adjacent months.

Figure 2 shows the annual "Distributor/Secondary" streets cycle from 2019 to 2022. A strong periodic drop in traffic activity can be observed on weekends. The traffic activity also captures special events like vacation and holiday times or the lockdown influence of the COVID-19 pandemic.

### 2.2.2 Vehicle Shares

We calculate the daily vehicle shares on different road types based on vehicle-specific counting time series and correct it with average weekday vehicle shares from the traffic model. A set of *valid* and *complete* time series were selected and grouped by vehicle class and road type. Daily median traffic volumes are computed for each group (e.g., median passenger car count of all counting stations located on a motorway). To acquire the vehicle share $\delta_{vc,r}$ for a specific road type, we divide the median count of individual vehicle types by the sum count. Figure 3 depicts the vehicle share time series on "Distributor/Secondary" streets in 2019. We apply additional correction factors because vehicle shares vary spatially, even on roads of the same type. For example, ring motorways have a higher HGV share than radial motorways (routes into the city center). The traffic model provides vehicle class-specific traffic volumes for HGV and LCV. They were used to infer average annual weekday shares

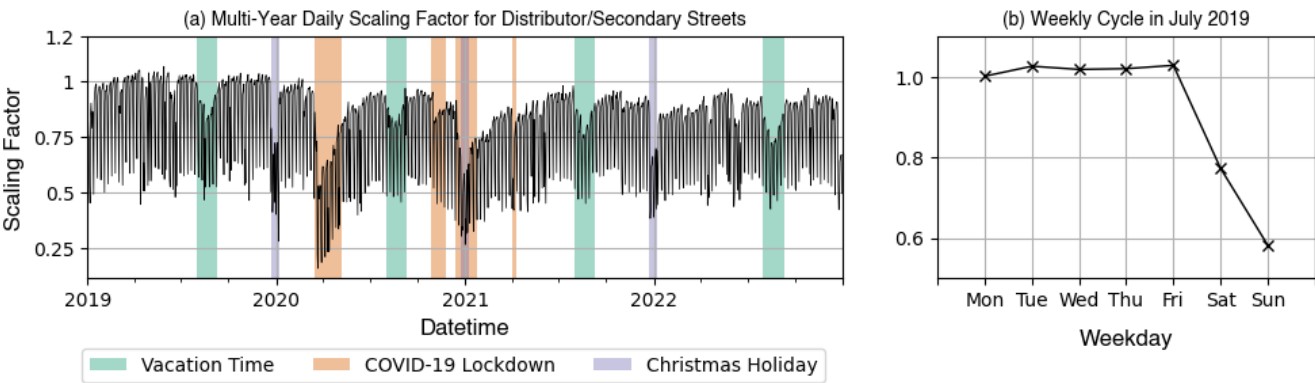

**Figure 2.** (a) Daily scaling factors derived from traffic counts for the years 2019 to 2022. The plot shows the normalized median count of all counting stations assigned to road links of the type "Distributor/Secondary". Each counting station's time series was normalized by its average 2019 norm-weekday count. (b) A weekly cycle in July 2019 displays the weekly pattern, showing consistent traffic from Monday to Friday and decreased activity on Saturday and Sunday.

**Table 2.** Formulas to calculate vehicle share correction factors $\kappa_{i,vc}$ based on vehicle shares derived from the traffic model and the traffic counters. These correction factors are used to incorporate spatial differences in vehicle shares.

| Vehicle Class | Correction Factor |
|---|---|
| **Heavy goods vehicle (HGV)** | $\kappa_{i,HGV} = \delta_{i,HGV}^{model} / \delta_{HGV,r}^{count,ref}$ |
| **Light cargo vehicle (LCV)** | $\kappa_{i,LCV} = \delta_{i,LCV}^{model} / \delta_{LCV,r}^{count,ref}$ |
| **Personal car (PC)** | |
| **Coach (BUS)** | $\kappa_i = \frac{1 - \kappa_{i,HGV} \cdot \delta_{HGV,r}^{count} - \kappa_{i,LCV} \cdot \delta_{LCV,r}^{count}}{1 - \delta_{HGV,r}^{count} - \delta_{LCV,r}^{count}}$ |
| **Motorcycles (MOT)** | |

$\kappa_{i,[HGV,LCV]}$ ... Vehicle share correction factor for HGV and LCV for the i-th road link.

$\kappa_i$ ... Vehicle share correction factor for all remaining vehicle classes for the i-th road link.

$\delta_{i,[HGV,LCV]}^{model}$ ... Average weekday vehicle share of LCV and HGV, derived from the traffic model for the i-th road lin,k.

$\delta_{[HGV,LCV],r}^{count,ref}$ ... Average weekday vehicle share of LCV and HGV, derived from the 2019 counting data on the respective road type.

$\delta_{[HGV,LCV],r}^{count}$ ... Vehicle share of LCV and HGV derived from daily counting data on the respective road type.

$\delta_{i,HGV}^{model}$ and $\delta_{i,LCV}^{model}$ for each road link. Subsequently, we calculate the average weekday share of the reference year 2019 based on the counting data $\delta_{HGV,r}^{count,ref}$ and $\delta_{LCV,r}^{count,ref}$ for each road type. The quotients $\kappa_{i,HGV}$ and $\kappa_{i,LCV}$ between the vehicle share in the traffic model and at the counting stations, aggregated by road category, were used to correct the vehicle share at each road link. The correction factor $\kappa_{i,vc}$ for the remaining vehicle categories is calculated by dividing the total share left after correcting for HGV and LCV by the original, uncorrected share for these categories. This factor ensures that the sum of all shares $\delta_{vc,r}$ equals one after correction. Table 2 shows the correction factors and formulas applied.

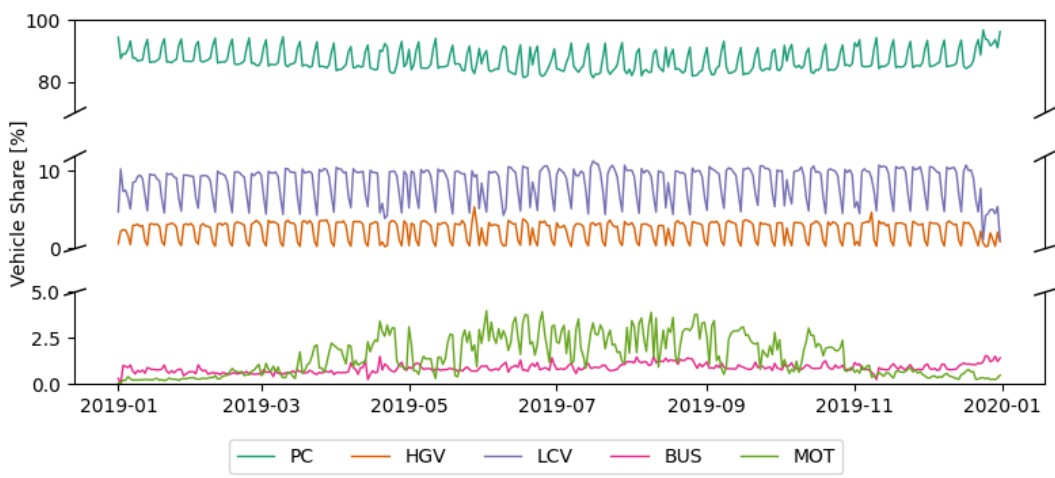

**Figure 3.** Vehicle share of all vehicle classes on distributor streets in 2019. PCs have the highest share of the total traffic, followed by LCVs, HGVs, MOTs, and Buses. A weekly pattern is observed for PC, LCV, and HGV, with an increase in the PC share and a decrease in the LCV and HGV share over the weekend. MOT shows an annual cycle, a trend that fits with the fact that motorbikes are more likely to be on the road on warm, sunny days. The share of BUS and MOT in total traffic is in the lower single-digit percentage range.

### 2.2.3 Diurnal Cycles

The daily cycle $\beta_{vc,d}$ is required to distribute the daily traffic volume to each hour. Diurnal patterns are similar for road and day types but differ for vehicle classes. The hourly traffic counts were normalized by the daily sum count and grouped by day type $d$ and vehicle class $vc$. For each month, a median cycle was calculated for each category. Figure 4 shows the diurnal cycle of PC on three different day types for 2019. Peaks during rush hours are pronounced in the weekday diurnal cycle. The Saturday and Sunday/Holiday cycles have a similar shape with a shift in time. Diurnal cycles for other vehicle classes can be found in appendix C1.

### 2.3 Emission Factor Selection

HBEFA distinguishes 365 different traffic situations by considering the road type, road gradient, speed limit, area type (rural vs. urban), and the level-of-service ($LOS$) (Notter et al., 2019). The road type, gradient, and speed limit are static for each road link, and the area type "urban" is used for the whole city area. The $LOS$ reflects the prevailing traffic condition and is estimated for each road link using the volume-capacity ratio $x_i$ (Eq. 2) and dedicated thresholds to distinguish between five classes: Freeflow, Heavy, Saturated, Stop&Go, and Stop&Go2 (gridlock with average speeds of 5-10 km/h). Hour capacity $C_i$ is an attribute of the traffic model. The vehicle-specific hourly traffic volume $q_{i,vc}$ is the product of the traffic model and the aforementioned temporal scaling factors (Eq. 1). The traffic volume for each vehicle class is converted into passenger car

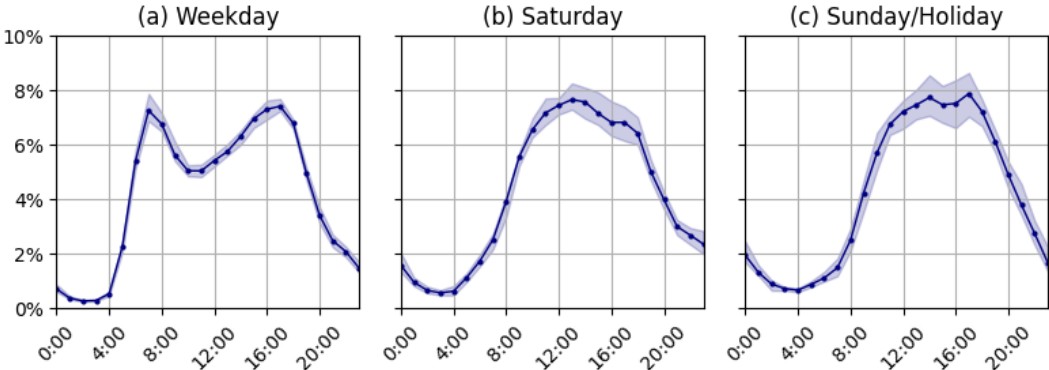

**Figure 4.** 2019 diurnal cycles of three different day types for PC traffic. The blue line represents the average cycle for all months of the year, while the blue-shaded area indicates the range within which the cycles of individual months fall. Individual cycles are computed for each vehicle class and month of the year.

equivalents (PCE) to account for differences in vehicle size and impact on traffic flow. This conversion enables the analysis of mixed traffic streams as if they consisted solely of passenger cars. The scaling factors $n_{vc}$ applied are shown in Table 3.

$$x_i = \frac{\sum\limits_{vc} q_{i,vc} \cdot n_{vc}}{C_i} \tag{2}$$

**Table 3.** Passenger Car Equivalent (PCE) scaling factors $n_{vc}$. These factors are applied to adjust the mixed traffic stream for the size and flow impact of different vehicle categories.

| vehicle class $vc$ | PCE factor $n_{vc}$ |
|---:|:---|
| PC | 1 |
| MOT | 1 |
| LCV | 1 |
| HGV | 2.5 |
| BUS | 1.75 |

Volume-capacity ratio thresholds to distinguish between $LOS$ classes cannot be universally applied. It is inevitable to validate and optimize them based on reference values. Schmaus et al. (2023) investigated the distribution of the vehicle kilometers traveled (VKT) among different $LOS$ classes in Germany based on floating car data. He shows distinct distributions for rural areas and agglomerations, whereby we employed the distribution for agglomerations in this study. Although this corresponds to a national average for urban areas, we assume that it reflects the situation in Munich well. We iteratively adjust the $VCR$ thresholds of different road types until the share of the VKT is within $1\%$ of the reference value. Further details of this

application-specific procedure and the targeted distribution on each road class can be found in the supplement and the related computational notebook. The resulting thresholds are shown in Table 4, and are further analyzed in Section 4.

**Table 4.** Optimized VCR-thresholds as applied in Munich to achieve the targeted distribution of VKT in different traffic conditions.

| Road Type | *Freeflow* | *Heavy* | *Saturated* | *Stop&Go* | *Stop&Go 2* |
|---|---|---|---|---|---|
| Reference (HBS, 2015) | 0.55 | 0.9 | 1 | >1 | - |
| Motorway National | 0.5 | 0.71 | 0.98 | 1.1 | > 1.1 |
| Primary-National | 0.33 | 0.5 | 0.7 | 1 | > 1 |
| Primary-City | 0.67 | 0.82 | 0.92 | 1.02 | > 1.02 |
| Distributor/Secondary | 0.37 | 0.5 | 0.63 | 0.8 | > 0.8 |
| Access/ Residential | 0.122 | 0.25 | 0.38 | 0.5 | > 0.5 |

## 2.4 Hot Vehicle Exhaust Emissions

The vehicle kilometers traveled (VKT) is calculated by multiplying the hourly, vehicle-specific traffic volume $q_{i,vc}$ with the geometric length of the road link $L_i$. Multiplying the VKT with a parameterized emission factor $EF^{hot}_{p,vc,r,s,v,LOS}$ results in the hot vehicle exhaust emission $E^{hot}_{i,p,vc}$ (Eq. 3). The emission factors are available for the average annual fleet composition of the corresponding vehicle class $vc$ and different pollutants $p$. Further parameters are the road type $r_i$, road gradient $s_i$, the maximum allowed speed $v_i$, and the level of service $los$.

$$E^{hot}_{i,p,vc} = q_{i,vc} \cdot EF^{hot}_{p,vc,r,s,v,LOS} \cdot L_i \tag{3}$$

## 2.5 Cold Start Excess Emissions

A "cold start" is the first minutes following the combustion engine's first start. The cold start period ends when the coolant reaches 343 K (70 °C) for the first time, but no later than five minutes after the initial engine start. (EC-JRC, 2017)

HBEFA provides emission surcharges $EF^{cold}_{p,vc,T}$ in grams per vehicle cold start differentiated by trip duration, length, and ambient temperature for passenger cars and light commercial vehicles. Trucks, buses, and motorcycles are therefore neglected in the following. Statistical averages were chosen for the duration and length of the trips, as this information is not available spatially resolved. A value-binned (-10°C, -5°C, 0°C, 5°C, 10°C, 15°C, 20°C, 25°C) hourly ambient temperature $T$ for Munich was derived from meteorology measurement data available in the city center. This measured temperature is not fully representative of every vehicle start in the study area, but it does provide a practical, time-resolved reference value in Munich for the application of the emission factors. Further influences, such as the parking location of the vehicle (e.g., underground garage, carport, street parking), cannot be examined in detail. The spatially distributed number of the vehicle starts $N_{vc}$ is available in the traffic model utilized in the study and was modulated using vehicle-specific temporal scaling factors for the road type "Distributor/Secondary."

The distance traveled and time taken for the engine to reach nominal temperatures are difficult to determine. So emissions are assigned to all roads except the motorway within a 1.5 km surrounding buffer. This radius represents how far vehicles can travel in the first 90 seconds at 60 km/h. A weighting factor was determined to distribute the emissions in relation to the traffic volume. This factor is calculated for each road link from the quotient of the traffic volume of the link and the sum of all traffic volumes of the links in the buffer zone. We assume a constant spatial distribution of cold start surcharges, as no separate information is available for days other than the average weekday of 2019.

$$E_{i,p,vc}^{cold} = N_{vc} \cdot EF_{p,vc,T}^{cold} \tag{4}$$

## 3 Results and Discussion

### 3.1 Annual Activity Data

Our proposed method combines a macroscopic traffic demand model with traffic counting data on different road types to achieve a data-based temporal scaling of traffic activity for emissions calculation. The primary result is an estimate of the vehicle kilometers traveled (VKT) for different vehicle classes and the respective traffic conditions. Figure 5a shows the annual VKT for 2019-2022. A 12.7 % decrease in the yearly VKT is present in 2020, which can be related to significantly reduced traffic activity due to the COVID-19 pandemic (Anke et al., 2021; Creutzig et al.). VKT increased by 1.5 % in 2021 and a further 3.5 % in 2022 compared to the previous year. The distribution of VKT across the different traffic conditions (LOS classes) has also changed over the years. In 2019, 53 % of the total vehicle kilometers traveled occurred under free-flow traffic conditions, which increased to 59 % in 2020. More kilometers have also been assigned to the traffic conditions *Stop&Go* and *Stop&Go 2* in 2019 compared to other years. In the HBEFA, the LOS class *Stop&Go* describes congestion with frequent stops and slow traffic, while *Stop&Go 2* stands for heavy congestion or traffic jams characterized by average speeds of 5-10 km/h. This indicates that traffic has become more fluid due to lower traffic volume in recent years.

The City of Munich's Department for Climate and Environmental Protection (RKU) bi-annually estimates the city's greenhouse gases. The most recent numbers are available for 2019 (Referat für Klima und Umweltschutz, 2022). On request, we received a more detailed evaluation, which splits emissions and the corresponding kilometers driven into total traffic and heavy goods traffic.

Their calculation is based on the BISKO Method (Bilanzierungs-Systematik Kommunal, eng.: Municipal Accounting System, IFEU (2019)) and is performed by analyzing the local final energy consumption. The emission factors for subsequent GHG calculation also include upstream emissions. Meaning that both the emissions caused by electricity production and upstream emissions from the fuel supply chain, such as fuel extraction, processing, and distribution, are included in the transportation sector (Well-to-wheel emissions) besides the sole activity-based emissions (Tank-to-wheel emissions).

The city uses traffic volume maps that represent the average weekday traffic on the major road network based on independent manual counts to calculate the activity. Minor roads are neglected. These volume maps are available for the total traffic and

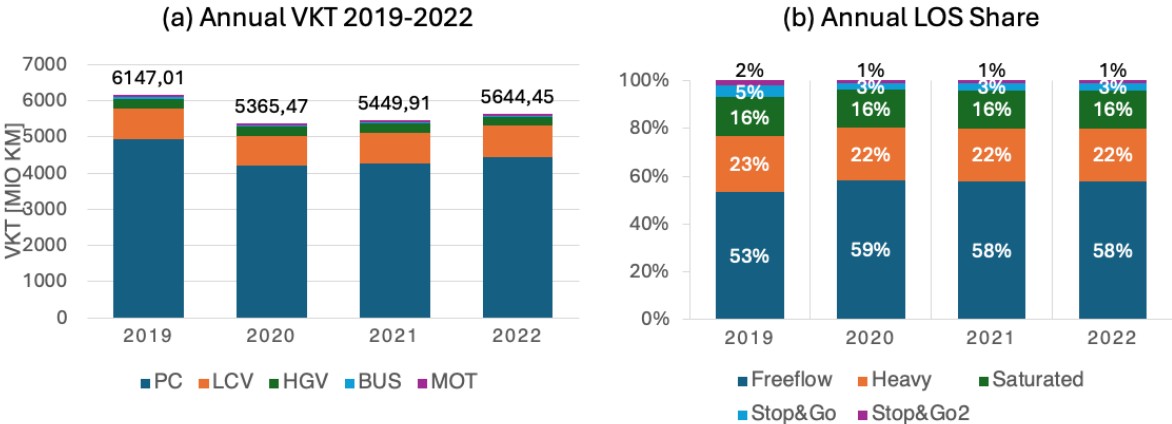

**Figure 5.** (a) depicts the total annual VKT in Munich for 2019-2022. A strong decrease due to COVID-19 is present in 2020. Since then, traffic activity has been growing but has not reached pre-COVID levels yet. (b) shows the relative distribution of the total VKT to different traffic conditions (LOS classes). The reduced traffic in 2020 resulted in more free-flow traffic and fewer kilometers traveled in highly congested conditions. The distribution is similar for 2020-2022.

heavy goods traffic. Table 5 compares the annual VKT of the RKU with the numbers calculated in the present study for 2019. Multiple vehicle classes were aggregated to represent the same vehicle categories as the RKU uses. To extrapolate the weekday traffic volume to an annual traffic volume, the RKU multiplies the value by 365 days and a factor of 0.8 to account for reduced weekend traffic. In comparison, our results show a 5.2 % lower VKT for heavy goods traffic, a 7 % higher VKT for the remaining vehicle classes, and a 6.4 % higher total VKT. This is related to the different temporal scaling methods, including minor roads in the calculation (which account for approx. 11 % of the total VKT on an average weekday) and the different input data used.

**Table 5.** Comparison of the vehicle kilometers traveled (VKT) between the official greenhouse gas reporting of the City of Munich's Department for Climate and Environmental (RKU) and the calculations in the present study. Our calculation results in a 6.4 % higher VKT estimate than the RKU, which is caused by different temporal scaling methods and input data.

| Vehicle Class | RKU [Mio. km] | DRIVE [Mio. km] | Difference |
|---|---|---|---|
| PC + LCV + MOT | 5424.42 | 5836.74 | + 7.06 % |
| HGV + BUS | 327.22 | 310.95 | - 5.23 % |
| SUM | 5751.64 | 6147.79 | + 6.44 % |

## 3.2 Annual Emission Estimates

Table 6 shows the annual total emission estimates for 2019 to 2022 and the year-to-year change. The decrease in all pollutants and GHGs in 2020 can be related to COVID-19, reduced traffic activity, and more fluid traffic, as shown above. Notably,

air pollutants decrease by a higher share than $CO_2$. This change can be attributed to the ongoing development of emissions legislation, which improves the emissions performance of the entire fleet (Gniffke et al., 2024).The growing proportion of emission-free vehicles also contributes to this, although, despite the exponential increase in the electric vehicle share in 2020, they only account for about 1.8 % of the annual vehicle kilometers driven in Germany (Knörr et al., 2023).

On a national level, the total annual $CO_2$ emission of the road transport sector decreased by -12.6 % in 2020, by -0.7 % in 2021, and increased by 2.4 % in 2022. Total $NO_x$ emissions decreased by -28.2 % in 2020, by -8.5 % in 2021, and by -5.5 % in 2022. For $CO$, the national authority reports a decrease of -23.7 % in 2020, a decrease of -3.2 % in 2021, and an increase of 4.3 % in 2022. (UBA, 2024) Similar trends are visible in the year-to-year change in Munich's annual emissions.

**Table 6.** Total traffic emission for $CO_2$ (fossil-fuel: $CO_{2,ff}$ , fossil- and bio-fuel: $CO_{2,ff+bf}$ ), $CO$, $NO_x$, $NO_2$, $PM$ and $CH_4$ in Munich. Total estimates and the year-to-year change from 2019 to 2022 are shown.

| Component | Unit | 2019 | | 2020 | | 2021 | | 2022 | |
|---|---|---|---|---|---|---|---|---|---|
| $CO_{2,ff}$ | kt | 1248.5 | - | 1050.1 | -15.9% | 1057.2 | 0.7% | 1065.2 | 0.8% |
| $CO_{2,ff+bf}$ | kt | 1312.0 | - | 1121.2 | -14.5% | 1129.7 | 0.8% | 1139.6 | 0.9% |
| $CO$ | t | 4194.0 | - | 3413.8 | -18.6% | 3303.2 | -3.2% | 3192.6 | -3.3% |
| $NO_x$ | t | 3433.6 | - | 2581.5 | -24.8% | 2411.4 | -6.6% | 2148.4 | -10.9% |
| $NO_2$ | t | 600.4 | - | 462.2 | -23,0% | 387.5 | -16.2% | 316.0 | -18.5% |
| $PM$ | t | 44.8 | - | 35.6 | -20.5% | 31,9 | -10.4% | 30.0 | -6% |
| $CH_4$ | t | 64.8 | - | 56.2 | -13.3% | 54,4 | -3.2% | 55.4 | 1.8% |

## 3.3 Spatial Distribution

Figure 6 depicts the spatial distribution of the emissions for different vehicle classes for 2019 as line sources along the road network. A dominant part (67 %) of the total $CO_2$ emissions comes from passenger car traffic, which can be seen in the pie chart (a). Further, 31 % can be attributed to light commercial and heavy goods vehicle traffic. In total, buses and motorcycle emissions constitute only 3 % of Munich's total $CO_2$ emissions. The spatial distributions are distinct and based on vehicle shares derived from the traffic model and the traffic counting data. The network of main roads is visible in the spatial distribution of the emissions.

In figure 7, we compare the spatial distribution of the total $CO_2$ emission with Germany's national inventory (referred to as UBA inventory) (Gniffke et al., 2024) and a European downscaled inventory (TNO GHGco v1.0) (Dellaert et al., 2019). Both use Germany's national estimates and disaggregate them with proprietary proxy maps. TNO utilizes open street map and open transport map-derived road networks and traffic volumes as well as population density. The UBA incorporates traffic volume as a proxy for the motorway and main roads. For further distribution to urban and rural roads, the UBA uses population density, number of registered vehicles per 1000 inhabitants, and spatial topology maps. Each inventory was gridded with a 1/60° latitude · 1/120° longitude (approximately 1 km · 1 km) grid, matching the original TNO grid's raster size and positioning. The cell values were normalized by the city total of each inventory to display them at a uniform scale.

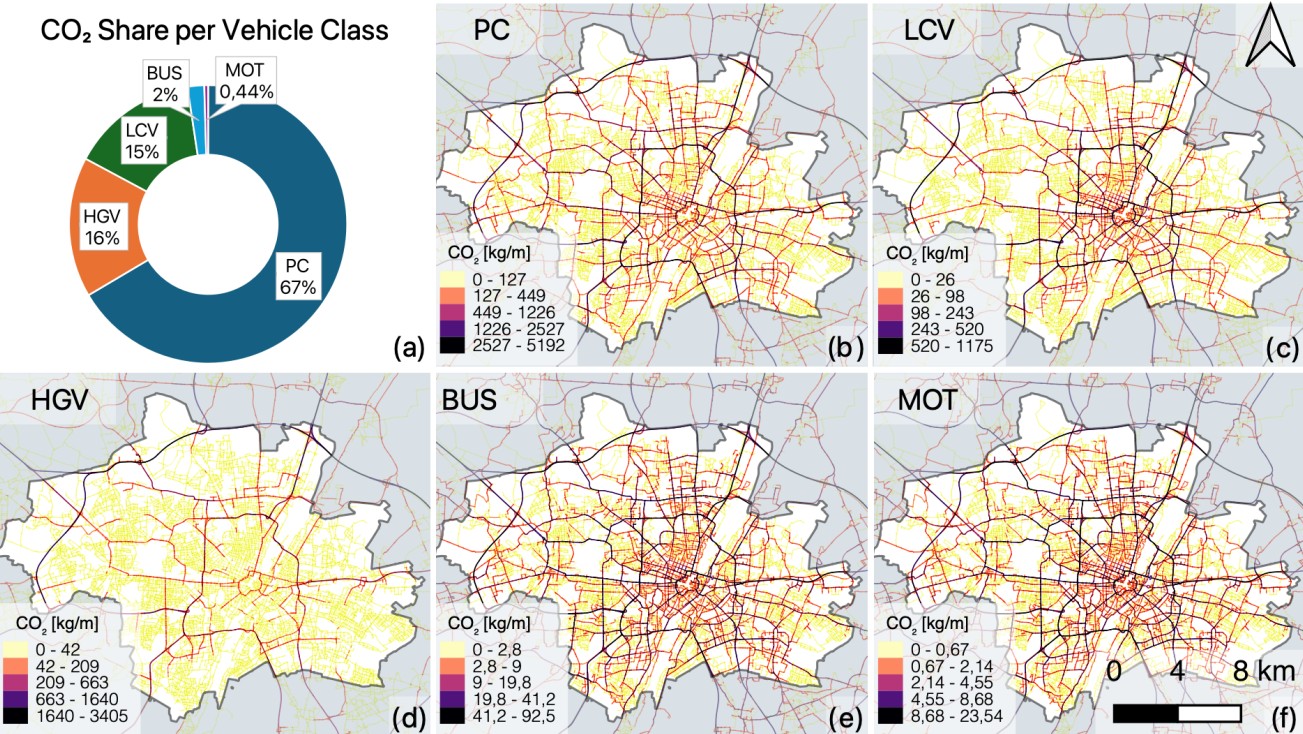

**Figure 6.** Spatial distribution and total share of $CO_{2,ff+bf}$ emissions of different vehicle classes in 2019. Passenger cars (PC) account for the largest share in emissions (67%). Followed by heavy goods vehicles (16%) and light commercial vehicles (15%). High emission values on the main roads are visible for all vehicle classes. While the emissions from HGV concentrate on main roads and the motorway, other vehicle classes emit on all roads. No distinct spatial patterns can be observed for BUS and MOT due to the absence of spatial information in the traffic model. In total, buses and motorcycles only constitute about 3% of Munich's emissions.

The first row of the maps shows the spatial distribution of the individual inventories. TNO allocates a large share of emissions to the city center, where the population density is highest. While the layout of the main roads is visible, it is not particularly pronounced. The UBA inventory shows a more uniform distribution across the urban area, and the main roads are slightly more pronounced. The second row depicts difference maps between DRIVE, UBA, and TNO. Both downscaled inventories (UBA and TNO) attribute lower emissions to the main road network than DRIVE, with TNO attributing significantly higher emissions 310 to the city center and UBA to some urban residential areas. Based on local activity data, the DRIVE inventory assigns about 90 % of the total $CO_2$ emissions to the main road network (motorway, primary, and secondary roads), reflecting traffic volumes and congestion. In conclusion, proxies such as population density or spatial topology maps are not very representative of the spatial distribution of traffic emissions on a city scale. Local activity data is essential for distributing traffic emissions at the city level.

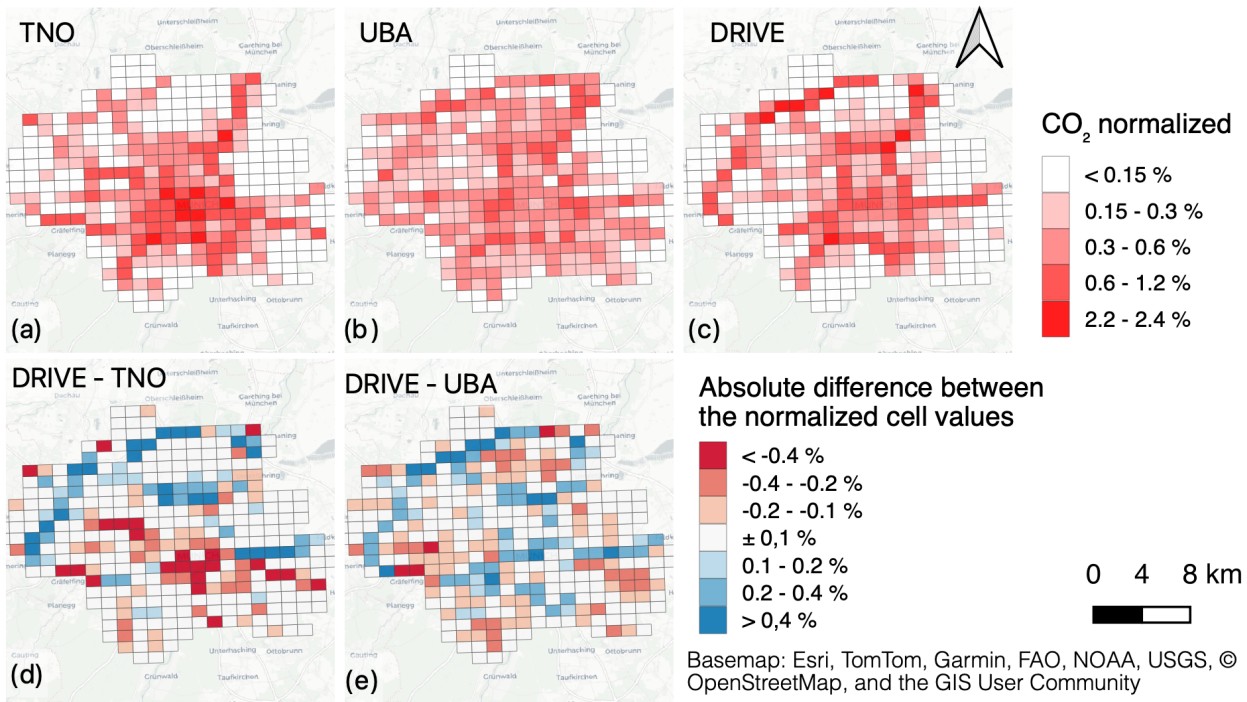

**Figure 7.** Comparison of the spatial distribution of $CO_2$ for the traffic sector from three inventory datasets. Plots (a)-(c) are normalized to the respective total values of the city to present them on a uniform scale. Therefore, each cell value represents the fraction of the total emissions attributed to this cell. TNO (a) attributes major emissions to the city center, i.e., the place with the highest population density. The UBA shows a more homogeneous spatial distribution, and compared to the DRIVE inventory, roads with high traffic volumes are less pronounced. The difference plots in (d) and (e) show the absolute difference between the normalized cell values. They indicate that UBA and TNO attribute lower emissions (blue) to parts of the main road network and higher emissions (red) to minor roads. DRIVE uses validated local traffic activity data, more accurately representing the spatial distribution of related traffic emissions. Both downscaled inventories reflect the incorporated spatial proxies.

Table 7 shows the emission totals for Munich and indicates large deviations between the datasets. The UBA attributes 47 % more while TNO attributes 24 % less $CO_2$ to the Munich region. The RKU reports 6.4 % less VKT but 6.2 % higher emissions, likely due to using a different emission factor database. For $NO_x$, the UBA shows 54 % higher values, while TNO indicates 14 % lower values. The $CO_{2,ff+bf}/NO_x$ ratios are very similar and range from 321 to 365, suggesting that the observed variations are due to differences in the underlying activity and proxy information rather than discrepancies in the emission factors.

For $CO$, the UBA attributes 287 % higher and TNO 83 % higher emissions. The $CO_{2,ff+bf}/CO$ ratio is 119 for UBA, 123 for TNO, and 297 for DRIVE and, therefore, cannot be attributed to different activity levels but emission factors. TNO and UBA distribute the same total emissions as reported by Germany. Consequently, the ratio between UBA and TNO falls within the

same range. In our estimation, we noticed that $CO$ emission factors in HBEFA are heavily dominated by petrol cars between Euro 4 and Euro 6ab. The emissions of these vehicles rise sharply at high speeds. Aggregated HBEFA emission factors for PC are 5 times higher for the road category "motorway" and 2.5 times higher for the area type "rural" compared to "urban". Accordingly, significant $CO$ emissions are expected primarily on the motorway. The motorway accounts for approximately one-third of the total VKT in this study, and 40% of the total hot exhaust emissions (total hot $CO$ = 2583 tons; motorway hot $CO$ = 1032 tons). Motorway-type road links in the traffic model used have a maximum speed limit of 120 km/h. In reality, however, the allowed speed is regulated depending on the traffic load, and on German motorways at free-flow conditions, no speed limit is applied. To further evaluate the impact of high free-flow speeds on the motorway, we applied the national, aggregated emission factor for motorways to all motorway road links in our study. This triples the $CO$ contribution from the motorway (motorway hot $CO$ = 3539 tons), resulting in a total $CO$ emission of 6701 t and a $CO_{2,ff+bf}/CO$ ratio 195.8. This suggests that we probably underestimate CO emissions on the highway, while methods based on proxies overestimate the urban share, where motorway speeds are generally lower due to high loads.

**Table 7.** Comparison of the total emission for fossil fuel $CO_2$ ($CO_{2,ff}$), fossil and biofuel $CO_2$ ($CO_{2,ff+bf}$), $CO$ and $NO_x$ from three different spatially explicit emission inventories in Munich (DRIVE, UBA, TNO). For this comparison, we selected the closest year available. The RKU estimate is the official number reported by the City of Munich and includes upstream emissions from the fuel supply chain (Scope 2). We compare the $CO_{2e,WTW}$ (Well-to-Wheel) emission in this case. All other emissions are tank-to-wheel, i.e., Scope 1 emissions. In addition, the table contains total values for $CO_{2,ff+bf}$ for subsets categorized according to the predominant road types in the grid cell. This suggests that UBA overestimates emissions at Secondary roads and TNO underestimates emissions on the Motorway.

| Component | Unit | DRIVE (2019) | UBA (2019) | | TNO (2018) | | RKU (2019) | |
|---|---|---|---|---|---|---|---|---|
| $CO_{2,ff}$ | kt | 1248 | - | - | 946 | - 24% | | |
| $CO_{2,ff+bf}$ | kt | 1312 | 1936 | + 47% | 993 | - 24% | | |
| $CO$ | t | 4194 | 16250 | + 287% | 7671 | + 83% | | |
| $NO_x$ | t | 3434 | 5299 | + 54% | 2946 | - 14% | | |
| $CO_{2e,WTW}$ | kt | 1499 | - | - | - | - | 1592 | + 6.2 % |
| $CO_{2,ff+bf}$, Motorway | kt | 513 | 551 | + 7% | 235 | - 54% | | |
| $CO_{2,ff+bf}$, Primary | kt | 220 | 263 | + 20% | 170 | - 23% | | |
| $CO_{2,ff+bf}$, Secondary | kt | 489 | 974 | + 99% | 489 | 0% | | |
| $CO_{2,ff+bf}$, Residential | kt | 27 | 94 | + 248% | 25 | - 7% | | |
| $CO_{2,ff+bf}$, None | kt | 0 | 53 | - | 27 | - | | |
| $CO_{2,ff+bf}/NO_x$ | | 363.4 | 365.4 | | 321.1 | | - | |
| $CO_{2,ff+bf}/CO$ | | 297.6 | 119.1 | | 123.3 | | - | |

## 3.4 Temporal Profiles

Temporal profiles were calculated to distribute the annual emissions over time and to calculate the hourly emissions. These profiles contain scaling factors for the annual average hourly emissions, where the mean value of all scaling factors is 1. Different profiles were calculated for hot emissions and cold start excess emissions. The temporal distribution of cold start emissions is significantly influenced by the ambient temperature, which is also reflected by the profile. Additionally, the temporal profiles are strongly correlated with the traffic volume. The hot exhaust profile captures the non-linear relationship between traffic volume and emission factors. Increased traffic volume leads to congestion, resulting in higher emissions. Figure 8 depicts the temporal profile for hot exhaust and cold start excess emissions of different pollutants for 2019. In addition, it shows an example of a week in June 2019. The morning and afternoon peak hours are visible on working days. By contrast, a lower scaling factor, i.e., lower emissions, is assigned on weekends and holidays. The temporal profile for cold-start excess emissions shows a more irregular pattern as it strongly depends on the ambient temperature. The $NO_x$ cold-start emission factor is negative for ambient temperatures above 25°C, which results in a negative scaling factor at the respective condition. This negative emission implies that $NO_x$ emissions are lower during the cold start phase of the vehicle than when the vehicle exhaust system is hot.

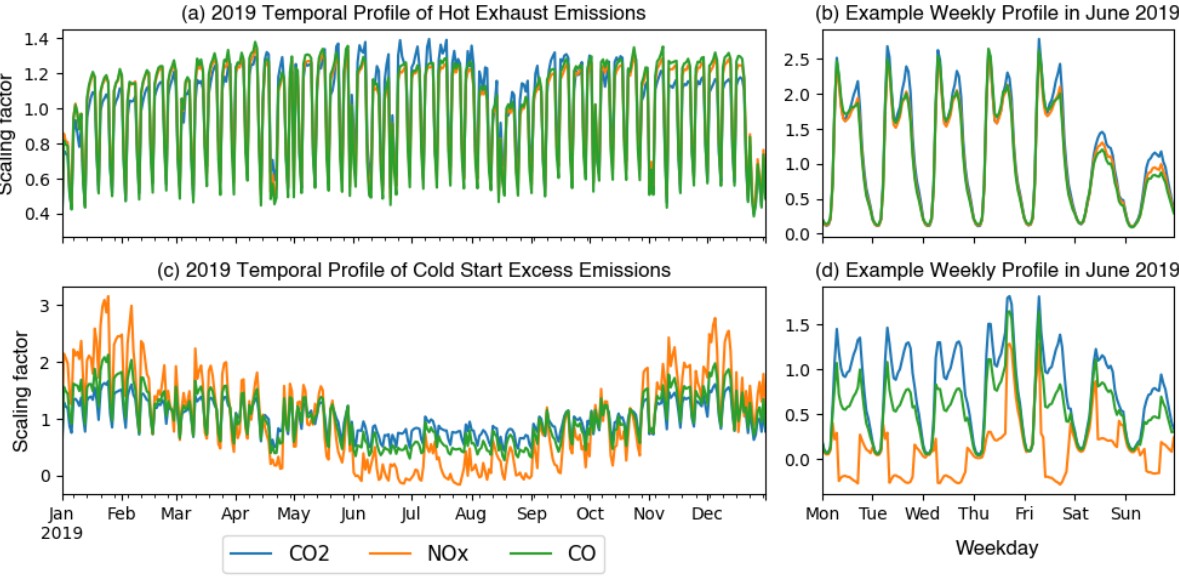

**Figure 8.** Temporal Scaling Factors for hot exhaust and cold-start excess emissions in 2019. Figure (a) illustrates the daily-averaged scaling factor for hot exhaust emissions, which closely follows traffic activity patterns. An example week in June 2019 (b) shows distinct rush-hour peaks during weekdays, while weekends generally have lower scaling factors due to reduced activity. Figures (c) and (d) illustrate the temporal profile of cold start excess emissions. Ambient temperature significantly affects this profile, leading to an annual cycle in the scaling factors. Negative cold start surcharges for NOx and NO2 are plausible, as these only represent a surcharge to the hot emissions. This means that in this case, the emissions during cold start are lower than the hot emissions. Overall, there are no negative emissions. The discontinuities observed in the weekly cycle arise from the non-linear, value-binned emission factors for cold start excess emissions used in HBEFA.

## 4 Sensitivity to Specific Model Parameters

The approach is subject to fine-tuning of parameters and some heuristic assumptions that can affect the final emissions result. In the following section, we will examine the sensitivity of the estimate to changes in the VCR thresholds and the cold start allocation radius.

### 4.1 Sensitivity Analysis of VCR Thresholds

In section 2.3, we propose to optimize the VCR thresholds to match the national distribution of traffic situations on each road type within $\pm 1\%$. This step is crucial for selecting the correct emission factor, as emissions increase sharply and non-linearly in congested traffic conditions. We tested a scenario with $\pm 10\%$ for all thresholds after the optimization, which is referred to as the nominal scenario. Table 8 shows the result of these scenarios and indicates that the emissions increase by 7 to 10 % if all VCR thresholds are lowered by 10 %, and the emissions decrease by 3 to 5 % if the thresholds are increased by 10 %. Raising the thresholds leads to an increase in free-flow conditions by 7 %, and lower Stop&Go conditions by 4 %. Lowering the thresholds results in a 7% decrease of VKT under free-flow conditions and a 6 % increase of Stop&Go conditions. We conclude that a change in the thresholds and the associated distribution of VKT across the traffic situations has a severe impact on the resulting emissions, and the optimization must be conducted with great care. The distribution of VKT based on city-specific statistics or the allocation of traffic conditions based on floating car data would further increase local representativeness.

**Table 8.** Emission and VKT-distribution sensitivity to a $\pm 10\%$ change of all VCR thresholds. The nominal scenario corresponds to the optimized threshold values used for the emissions calculation and shown in Table 4.

| | Emissions | | | Traffic Situations | | | | |
|---|---|---|---|---|---|---|---|---|
| | $CO_2$ [kt] | $CO$ [t] | $NO_x$ [t] | Freeflow | Heavy | Satur. | St&Go | St&Go2 |
| **Nominal Scenario** | 1287 | 2583 | 3333 | 53.4% | 22.5% | 16.3% | 5.6% | 2.2% |
| **Thresholds -10%** | 1403 | 2760 | 3664 | 46.6% | 21.7% | 17.6% | 8.2% | 6.0% |
| *rel. change* | *+ 9.0%* | *+ 6,9%* | *+ 9,9%* | *- 6.8%* | *- 0.9%* | *+ 1.3%* | *+ 2.7%* | *+ 3.7%* |
| **Thresholds +10%** | 1226 | 2519 | 3153 | 60.1% | 22.0% | 14.2% | 2.7% | 1.0% |
| *rel. change* | *- 4.7%* | *- 2.5%* | *- 5.4%* | *+ 6.7%* | *- 0.5%* | *- 2.1%* | *- 2.9%* | *- 1.2%* |

### 4.2 Sensitivity Analysis of Cold-Start Allocation Radius

The number of vehicle starts is distributed across spatial zones in the traffic model and available for PC and LCV. We assign vehicle starts to all intersecting road links within the zone and a surrounding 1.5 km buffer radius, weighted by the traffic volume of the respective road link. Motorways and primary roads are generally excluded. To test the influence of the allocation radius, two additional scenarios with a 0.8 km and a 2 km buffer radius were tested. A study by Pina and Tchepel (2023) shows a typical driving distance of 5 km for inner-city journeys under cold start conditions, with excess emissions being highest at

the start of the journey and then decreasing exponentially. We conclude that changing the allocation radius does not change the total emission at a policy-relevant level. Lowering the buffer radius generally leads to an increase in cold starts, attributed to residential roads as shown in Table 9. Figure 9 shows a difference map between allocated cold start emissions of the nominal scenario and 800 m and 2 km buffer radius, respectively. Larger differences are visible outside the city center, particularly for 800 m scenario. However, neither map shows a systematic correlation between the buffer radius and spatial distribution that

could indicate an inadequate assumption, and the buffer distance has little influence on the city's total. The 1.5 km radius is applied until more conclusive information becomes available.

**Table 9.** Sensitivity of results when changing the cold start buffer radius. Total emissions change by less than 1 %.

| | Total Emissions | | | Road Types [starts/day] | |
|---|---|---|---|---|---|
| | CO2 [kt] | NOx [t] | CO [t] | Secondary | Residential |
| **Buffer = 1.5 km** | 22.46 | 64.16 | 1549.23 | 2127309 | 806793 |
| **Buffer = 0.8 km** | 22.33 | 63.77 | 153975 | 2106709 | 827393 |
| *rel. change* | - 0.6 % | - 0.6 % | - 0.6 % | - 1.0 % | + 2.6 % |
| **Buffer = 2 km** | 22.67 | 64.75 | 1563.39 | 2137269 | 796834 |
| *rel. change* | + 0.9 % | + 0.9 % | + 0.9 % | + 0.5 % | - 1.2 % |

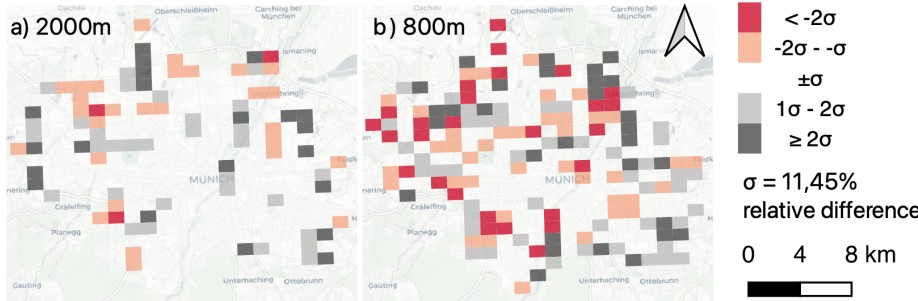

**Figure 9.** Relative difference in total cold start emission surcharges with an allocation buffer radius of a) 2 km and b) 800 m on a 1x1 km grid cell level. Larger relative differences can be observed outside the city center. However, no further systematic correlation between buffer radius and spatial distribution can be observed.

## 5   Uncertainty Analysis

### 5.1   Uncertainty of the Activity Data

The absolute uncertainty of the activity data $\Delta AD$ can be calculated by subtracting the modeled traffic volume from actual

traffic counts measured by the traffic detector on the respective road link. Accordingly, the relative uncertainty $\frac{\Delta AD}{AD}$ is obtained

by dividing the absolute error by the actual value, i.e., the counted traffic volume. This assessment was performed at an hourly, daily, and annual temporal aggregation, using the 2019 counting data from valid-flagged counting stations assigned to 82 individual road links. This traffic counting data is statistically independent as it was not used to calibrate the traffic model in the first place. Instead, the city uses manual counts and data from Germany's national mobility study MID (Mobilität in Deutschland (Nobis and Kuhnimhof, 2018)).

Figure 10 shows the analysis result. A systematic, overall positive bias can be observed for the hourly, daily, and annual traffic volume (Fig. 10 b, d, and f). Counting stations on Distributor/Secondary roads tend to show higher values, while the model slightly overestimates the traffic volume on primary city roads and the motorway with high traffic volumes. However, it is also possible that the traffic counting stations, which are taken as the ground truth in this analysis, underestimate the volume of traffic, particularly at high volumes, e.g., due to incorrect or missing counts, or the malfunctioning of individual detectors. Lu et al. (2008) discusses loop fault detection and correction algorithms, but they require traffic data with higher temporal resolution than we use. In congested traffic situations, a combination of floating car data and traffic loop detector information shows superior performance in accurately estimating traffic variables compared to only using one data source (Qiu et al., 2010). Floating car data, however, is not available in our case. The counting data was assumed to be the ground truth to the best of our knowledge, and the effects mentioned were not examined in detail.

The 95 % confidence interval (95 % CI) of the relative deviation between detector and model values is (-52%, +110%) for hour values, (-32%, +64%) for daily values, and (-23%, +52%) for annual values. This shows a significant reduction in uncertainty with increasing temporal aggregation.

## 5.2 Uncertainty of the Emission Factor

Allekotte et al. (2023) examines the relative uncertainties of HBEFA emission factors $\frac{\Delta EF}{EF}$ for air pollutants nationally. The authors calculate uncertainties using Monte Carlo simulations. On average, a lower limit of -55 % to -10 % and an upper limit of +10 % to +90 % for regulated and non-regulated air pollutants was determined. However, the emission factors in the study are not based on the same granular activity rates as in HBEFA (*g/km*) but are expressed in *g/MJ* energy consumption. The emission factors were also aggregated according to different categories and distinguished between vehicle classes, pollutants, drive-train technologies, emission concepts, and road categories. Nevertheless, the uncertainties stated in this report were used because, to our current knowledge, this is the only information available on the uncertainty of HBEFA emission factors in Germany. Median upper and lower bounds for $NO_x$, $CO$, and $PM_{2.5}$ for the dominant vehicle categories PC, LCV, and HGV were extracted from the report. A weighted average was calculated based on the respective vehicle classes' total emissions. The report does not contain uncertainties for $CO_2$, as the calculation of $CO_2$ is based on energy source sales, and the approach for estimating its uncertainty is not model-based. On the national level, the 95 % CI for $CO_2$ ranges from $\pm 0.4\%$ for gasoline to $\pm 5\%$ for biodiesel. A conservative estimate of $\pm 5\%$ for all fuel and vehicle types is assumed. Table 10 shows the emission factor uncertainties and distributions applied. Identical to Super et al. (2024), we assume a log-normal distribution and calculate the equivalent Gaussian standard deviation $\frac{\Delta EF}{EF}$ using equation 5.

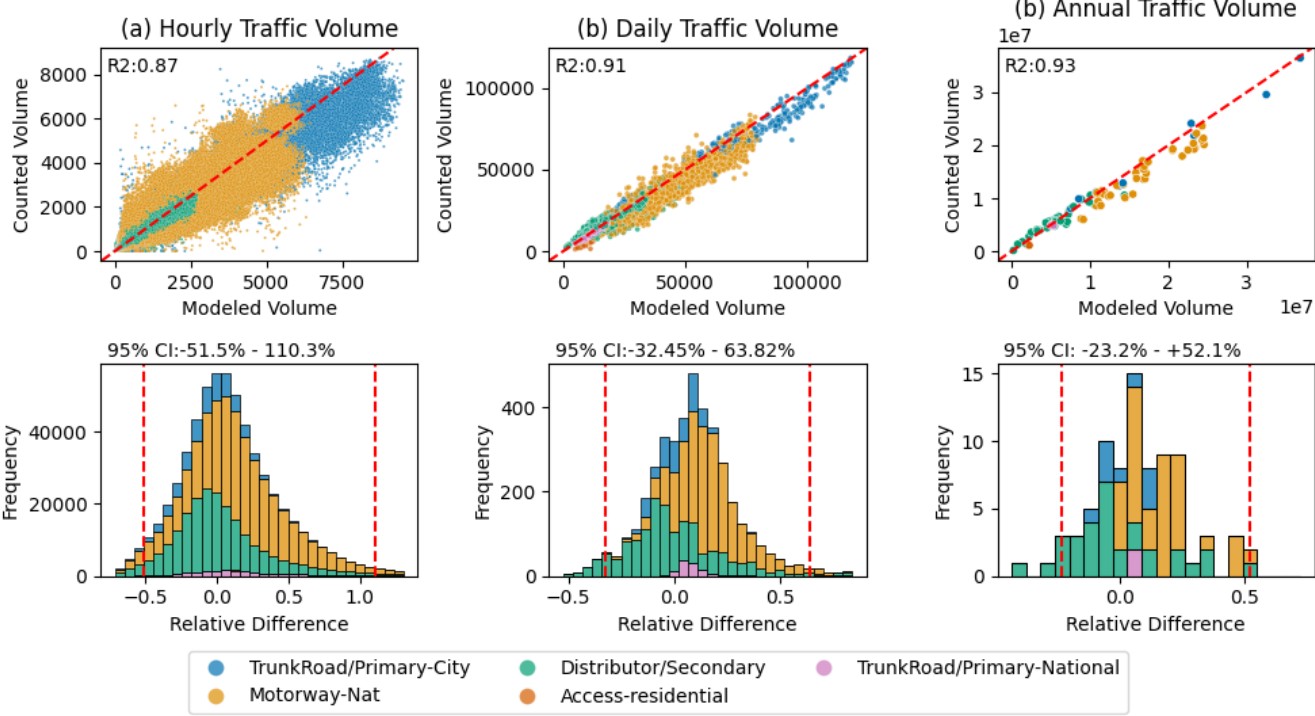

**Figure 10.** Uncertainty assessment of the activity data used in the present study: (a) and (b) show how hourly measured traffic counts match the modeled traffic volume for the total traffic at 82 road links classified by the road type. (c) and (d) illustrate the same for daily and (e,f) for annual traffic volumes. The model seems to overestimate the traffic volume at higher levels (Motorway and TrunkRoad/Primary-City) and underestimate the traffic at lower-level roads (Distributor/Secondary). The 95 % confidence interval of the relative uncertainty significantly decreases with temporal aggregation, as expected. Considering the large extent of the traffic model and the high number of counting stations, these numbers can be well accepted.

$$\frac{\Delta EF}{EF} = \frac{ln(1 + |lim_{upper}|) - ln(1 - |lim_{lower}|)}{4} \tag{5}$$

## 5.3 Uncertainty of the Emission Estimate

The traffic activity is used to estimate the traffic condition (Sec. 2.3), which is a fundamental parameter in selecting the emission factor. Therefore, traffic activity and the emission factor are correlated. This correlation, however, is not linear and cannot be determined analytically. We use the counted traffic volume and resulting volume-capacity ratio to calculate the emissions directly at the detector locations for methodological and computational simplicity. We then subtract the modeled emission of the respective road to calculate the absolute emission uncertainty $\Delta E^*$ and the relative uncertainty $\frac{\Delta E^*}{E^*}$ respectively. If the modeled and counted traffic volumes deviate so much from each other that different traffic conditions are estimated, this

**Table 10.** Emission factor uncertainties and distributions for air pollutants and carbon dioxide. The values were collected from the Allekotte et al. (2023) assessment of the uncertainties in HBEFA emission factors. The upper bound $lim_{upper}$ and lower bound $lim_{lower}$ were collected, and a $\frac{\Delta EF}{EF}$ for an equivalent Gaussian distribution was calculated.

| Component | $lim_{lower}$ | $lim_{upper}$ | Distribution | $\frac{\Delta EF}{EF}$ |
|---|---|---|---|---|
| $NO_x$ | - 25 % | + 31 % | log-normal | 13.9 % |
| $CO$ | - 33 % | + 43 % | log-normal | 19.0 % |
| $CO_{2,ff}$ | - 5 % | + 5 % | normal | 2.5 % |

leads to significant variations in emissions. Therefore, the inclusion of data from multiple traffic counting stations inherently introduces high variability to the estimated emissions. Emissions are overestimated in some areas and underestimated in others. Finally, the emission factor uncertainty $\frac{\Delta EF}{EF}$ is added to calculate the total emission uncertainty. The total emission uncertainty 425 $\frac{\Delta E}{E}$ is approximated using equation 6.

$$\frac{\Delta E}{E} \approx \sqrt{\left(\frac{\Delta E^*}{E^*}\right)^2 + \left(\frac{\Delta EF}{EF}\right)^2} \tag{6}$$

The relative uncertainty of the emission estimate $\frac{\Delta E^*}{E^*}$, excluding the emission factor uncertainty, was calculated for hourly, daily, and annually aggregated emissions. A log-normal distribution with zero mean was fitted to the values within the 95% CI. Figure 11 shows the histogram and equivalent standard deviation of the relative emission uncertainty. We estimate 33.1 430 % - 36.7 % for hourly and 27.9 % - 28.0 % for daily values, depending on the pollutant. It shrinks from hourly to daily deviations for all three components. For annual aggregated values, the standard deviation increases to 26.0 % -29.4 % due to the greater significance of individual stations with strong deviations, i.e., systematic bias. If the detector value deviates significantly from the model value, then different traffic situations are estimated, which leads to a systematic deviation in the calculated emissions.

Table 11 presents the total emission uncertainty $\frac{\Delta EF}{EF}$. The investigation focused on road links where traffic detectors were installed, ranging from short segments in densely interconnected urban areas to longer stretches on highways. We assume that emissions within 100 x 100 m grid cells exhibit identical levels of uncertainty due to a high spatial error correlation.

**Table 11.** Total Emission uncertainty $\frac{\Delta E}{E}$ at a road link level for $CO_2$, $NO_x$ and $CO$. Three different values for different temporal aggregations are shown. Due to strong spatial error correlation, we assume identical uncertainty levels for emissions within 100 x 100 m grid cells.

| Component | Hourly Uncertainty | Daily Uncertainty | Annual Uncertainty |
|---|---|---|---|
| $NO_x$ | 37.1 % | 31.9 % | 33.9 % |
| $CO$ | 41.4 % | 33.8 % | 32.4 % |
| $CO_{2,ff}$ | 34.2 % | 28.0 % | 29.0 % |

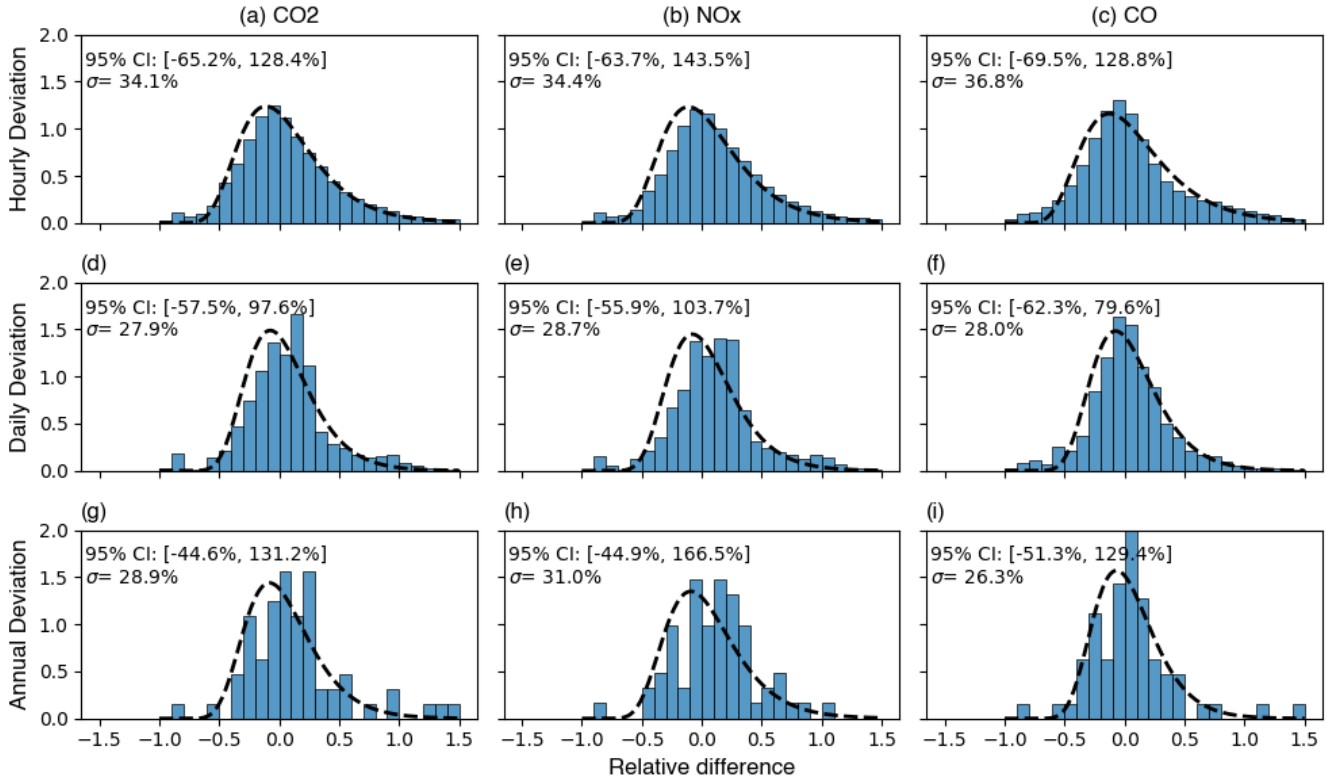

**Figure 11.** The estimated uncertainties $\frac{\Delta E^*}{E^*}$ for $CO_2$, $NO_x$, and $CO$ emissions were calculated using traffic volumes and congestion levels from both modeled and counted data at 64 detector locations. The difference between model and detector emissions is shown for hourly (a-c), daily (d-f), and annual (g-i) aggregation levels. There is a noticeable reduction in relative uncertainty when moving from hourly to daily aggregation. However, when moving from daily to annual aggregation, the calculated values increase due to the higher significance of individual stations with strong deviations, resulting in less reliable statistics. A log-normal distribution with a zero mean is fitted to the data within the 95 % confidence interval, and the equivalent standard deviation was calculated.

## 5.4 Limitations of the Uncertainty Assessment

The uncertainty analysis focuses only on hot vehicle exhaust emissions and does not consider cold start emissions due to the lack of comparable data. This approach is adequate for $CO_2$ and $NO_x$, as these emissions are mainly generated when the engine is hot. For $CO$, strongly influenced by excess emissions during cold starts, the analysis likely underestimates the uncertainty because cold start emissions are more uncertain than hot emissions. Furthermore, on a city level, no specific information is available regarding the fleet composition, such as powertrain technologies and emission concepts, so statistical averages provided in HBEFA are used. These factors can vary significantly based on vehicle type, age, maintenance, and operating conditions, which may not be fully represented in a generalized dataset. Moreover, estimating traffic conditions using the volume capacity ratio is a simple, robust, and scalable method, yet it is not very accurate in urban road networks. The

traffic flow is more often limited by the capacity of intersections than by the road links between them. The optimization applied (section 2.3) allows us to achieve a representative distribution of traffic conditions for the whole city on an annual average. However, we can not explicitly account for congestion effects such as queues and spillbacks. Despite these limitations, we assume the volume capacity ratio provides a reasonably accurate estimate of traffic conditions on the road link. But, at a road link level, congestion may introduce more uncertainty than reported. Finally, we do not explicitly take the correlation between traffic activity and the emission factor into account. If the traffic activity is estimated inaccurately, it leads to an incorrect traffic condition, and subsequently, a wrong emission factor is applied. A sensitivity analysis quantifying the impact of this correlation could further clarify its influence. The level of uncertainty also exhibits a daily pattern: at night, when traffic activity is low, the likelihood of a traffic jam is also low. However, during the day, especially during peak hours, the chances of experiencing traffic jams increase significantly. In future work, conducting a Monte Carlo simulation that incorporates the uncertainties related to traffic activity and emission factors during specific time periods could enable a probabilistic representation of how uncertainties propagate and better quantify the uncertainty of the emissions estimate.

## 6    Conclusions

We developed a road traffic emission inventory framework based on a macroscopic traffic model (static traffic model) and data from multiple traffic counting stations (dynamic traffic data). We utilize the counting data to extrapolate the traffic model temporally, increase the temporal resolution to one hour, and calculate hourly vehicle shares on different road types. The counted vehicle shares are then further adjusted using traffic model-based vehicle shares to account for spatial differences. In this way, we effectively combine the spatial information from the traffic model with the temporal information from the count data.

Based on the congestion level, road type, speed limit, road gradient, and vehicle class, the model selects the corresponding emission factor from the HBEFA 4.2 database. The congestion level is estimated using volume-capacity ratio thresholds optimized to reflect the national average distribution of traffic situations (Schmaus et al., 2023). By multiplying the emission factors by the vehicle-specific traffic volume and length of the road link, we achieve hourly hot exhaust emission estimates of different pollutants. To account for cold start excess emissions, the number of vehicle starts, which is included in the traffic model, was scaled using an average traffic activity profile. The hourly, local ambient temperature, averaged trip lengths, and parking durations were used to select emission factors for cold start emission calculations. Emission estimates are presented for 2019-2022.

In 2019, we estimate 6.2 % lower emissions compared to the city's assessment. However, we found significant differences compared to other spatially explicit, downscaled emission inventories. The German national inventory from UBA denotes 47 % higher, and TNO's European inventory 24 % lower $CO_2$ emissions to Munich. These proxy-based downscaling methods tend to underestimate traffic emissions along the leading road network and place a stronger emphasis on residential areas or places with high population density. This issue arises from the proxies used, which are not very representative of traffic

emissions on an urban scale. Using local activity data and conducting a bottom-up estimation is essential to create an accurate
emission map.

Finally, we assess the uncertainty of our estimates at a road-link-level resolution. The uncertainty arises from both the activity data and the emission factor. To assess the uncertainty in activity data, we compared the modeled traffic volume with actual counts from traffic counting stations. Our analysis demonstrates that uncertainty decreases as temporal aggregation increases. On an hourly basis, the 95 % confidence interval ($\pm 2\sigma$) is [-52 %, +110 %]; on a daily basis, it is [-32 %, +64 %]; and on an
annual basis, it is [-23 %, +52 %]. Subsequently, we calculated the total emission uncertainty by estimating emissions using the counting data and comparing this estimate to the modeled emission of the respective road. The uncertainty, which we obtained from Allekotte et al. (2023), was added to this estimate. Total emission uncertainties ($\sigma$) range between 28 % to 41 % depending on the temporal aggregation level (hourly, daily, annual) and pollutant.

Emission inventories with high spatial and temporal resolution are fundamental for atmospheric modeling in urban areas.
In particular, accurate temporal profiles are required to capture the temporal dynamics in high-resolution models, which is a limiting factor when using standard temporal profiles (Berchet et al., 2017). Our comprehensive use of counting data for vehicle class-specific temporal extrapolation has the potential to greatly improve the accuracy of atmospheric inversion in urban areas. The proposed framework effectively captures special events, such as COVID lockdowns, vacation periods, and individual holidays, and can also be adapted for near-real-time applications.

We see a lot of potential to further enhance the framework's capabilities by implementing methods to more accurately model the level and the location of congestion. While this is expected to change the result only modestly on the city scale, particular and localized emission hotspots are expected, which could inform policymakers where to target mitigation efforts (Tsanakas et al., 2020; Gately et al., 2017). Future work could include floating car data from TomTom or INRIX, providing actual vehicle speed information to more accurately model traffic congestion. Traffic congestion indexes, as used by Li et al. (2023), or microscopic
traffic models that provide detailed movement information on individual vehicles, are another possible implementation in this regard.

A limitation to the application of the framework is the availability of data in the target city. The model requires a macroscopic traffic model and vehicle-specific counting data for its calculations. However, in well-developed cities, both data sets are usually available and can be requested from the city administration if they are not publicly accessible. Additionally, the HBEFA
emission factor database is specifically tailored for European cities and is best applied in countries directly supported by the database, such as Germany, Switzerland, Austria, France, Sweden, and Norway. HBEFA emission factors are based on national vehicle fleets, driving patterns, and fuel characteristics, which may not fully represent conditions in other regions. Still, users can also create custom fleet compositions representative of other countries, although this falls outside the scope of the presented methodology.

Despite the limitations in accurately modeling traffic conditions and the limited knowledge of the local fleet composition, the proposed method provides a comprehensive, data-driven, and scalable approach to exploit static travel demand models and counting data from multiple traffic counting stations to estimate road transport emissions and their uncertainty. In future work, we will combine the transport inventory with high-resolution emissions data from other sectors such as heating, industry and

public heat, power generation, and human respiration. This complete inventory can then be used in atmospheric modeling
systems and optimized using the observations available in Munich.

*Code and data availability.* A code repository with detailed documentation and publicly available data sets is made available at https://doi.org/10.5281/zenodo.14644298 (Kühbacher et al., 2025). Updates and later versions can be accessed under https://github.com/tum-esm/drive-inventory

*Author contributions.* DK designed and implemented the framework, performed all analysis, and wrote the original draft of the paper. JC
provided supervision throughout the whole project. PA provided additional supervision at the early project phase. JC, PA, IS, and HDG
provided methodological and scientific input and contributed to the analysis of the results at all stages of the project. MI provided scientific
input on traffic engineering and contributed to the analysis of traffic-related results. The project was originally conceived by JC and HDG.
All co-authors reviewed the manuscript and provided feedback.

*Competing interests.* The authors declare no competing interest.

*Acknowledgements.* We thank the City of Munich (Dr. Carlos Llorca Garcia, Verena Hartlieb and Wolfgang Qual) for data provision, insightful exchanges and general support of the topic. We also thank Stefan Feigenspan (Umweltbundesamt) for providing data and insights
on Germany's national inventory. Further thanks go to Benedict Notter (INFRAS), who supported the comprehensive implementation of
HBEFA, provided feedback on results and methodological suggestions. The authors also thank the IOCS Cities project and members who
enabled the project. Finally, we thank Andreas Luther who reviewed the text.
For grammar and spell checking, DeepL and Grammarly were used.

*Financial support.* This work has been funded by the ICOS PAUL project: PAUL, Pilot Applications in Urban Landscapes - Towards integrated city observatories for greenhouse gases (ICOS Cities), funded by the European Union's Horizon 2020 Research and Innovation Programme under grant agreement No 101037319. Furthermore, the work is partly supported by the ERC consolidator grant CoSense4Climate
(Grant 101089203).

## 535  Appendix A:  List of Symbols

| | |
|---|---|
| $E_{i,p,vc}^{hot}$ | Hot exhaust emission for pollutant $p$ of vehicle class $vc$ on the i-th road link. 11 |
| $EF_{p,vc,r,s,v,LOS}^{hot}$ | Parameterized emission factor for hot exhaust emission calculation [g/km]. 11 |
| $E_{i,p,vc}^{cold}$ | Cold-start excess emission of pollutant $p$ and vehicle class $vc$ on the i-th road link. 12 |
| $EF_{p,vc,T}^{cold}$ | Parameterized emission factor for excess emission calculation [g/vehicle start]. 11, 12 |
| $T$ | Ambient temperature [°C]. 11 |
| $p$ | Index for different pollutants (e.g. $CO_2$, $NO_x$). 11 |
| $d$ | Day type (weekday, Saturday, Sunday/Holiday). 7 |
| $vc$ | Vehicle class: PC, LCV, HGV, MOT, BUS. 11 |
| $i$ | Index for different road links. 5 |
| $s_i$ | Slope of the road [%]. 5, 11 |
| $L_i$ | Road length of the i-th road link [km]. 11 |
| $C_i$ | Hourly vehicle capacity of the i-th road link [veh/h]. 5, 9, 10 |
| $v_i$ | Allowed speed of the i-th road link [km/h]. 5, 11 |
| $r_i$ | Road type of the i-th road link. 5, 6, 11 |
| $q_{i,vc}^{model}$ | Modeled average annual weekday traffic flow of the i-th road link [veh/day]. 5, 7 |
| $\delta_{i,HGV}^{model}$ | Heavy goods vehicle (HGV) share of the i-th road link in the traffic model. 8 |
| $\delta_{i,LCV}^{model}$ | Light cargo vehicle (LCV) share of the i-th road link in the traffic model. 8 |
| $q_{i,vc}^{count,ref}$ | Counted average annual weekday traffic flow of the i-th road link, if and counting station is assigned [veh/day]. 6, 7 |
| $q_{i,vc}$ | Vehicle-specific hourly traffic flow of the i-th road link [veh/h]. 7, 9, 10, 11 |
| $\alpha_r$ | Daily, road type specific scaling factor (annual cycle). 7 |
| $\beta_{vc,d}$ | Hourly, road type, and vehicle class specific scaling factor (diurnal cycle). 7 |
| $\delta_{vc,r}$ | Daily road-type and vehicle class specific share (vehicle share). 7, 8 |
| $\kappa_{i,vc}$ | Vehicle share correction factor to account for spatial differences in the vehicle share among a single road type. 7, 8 |
| $x_i$ | Traffic volume to capacity ratio. 9, 10 |
| $n_{vc}$ | Passenger car equivalent of the vehicle class $vc$. 10 |
| $N_{vc}$ | Number of vehicle starts of the vehicle class $vc$. 11, 12 |
| $\delta_{HGV,r}^{count,ref}$ | Heavy goods vehicle (HGV) share based on the 2019 average weekday counting data of the respective road type. 8 |
| $\delta_{LCV,r}^{count,ref}$ | Light cargo vehicle (LCV) share based on the 2019 average weekday counting data of the respective road type. 8 |
| $\frac{\Delta AD}{AD}$ | Relative uncertainty of the activity data. 20 |
| $\frac{\Delta EF}{EF}$ | Relative uncertainty of the emission factor. 21, 22, 23 |
| $\frac{\Delta E^*}{E^*}$ | Relative uncertainty of the emission estimate without emission factor uncertainty. 22, 23, 24 |
| $\frac{\Delta E}{E}$ | Relative uncertainty of the emission estimate including the emission factor uncertainty. 23 |

# Appendix B: Counting Data Model and Model Parameters

**Table B1.** Data model for traffic counting data from different sources. Data following this structure can be processed by the modules implemented in the framework.

|  | Column | Data Type | Description |
|---|---|---|---|
| **PK** | *index* | int | Unique index for each row of the dataset |
| **FK** | *road_link_id* | int | Assigns the data row to a road link in the traffic model |
|  | *date* | timestamp | Date of the counting information |
|  | *vehicle_class* | str | Vehicle class of the counting information |
|  | *road_type* | str | Road type of the corresponding road segment |
|  | *day_type* | int | 0: norm-weekday, 1: weekday, 2: Saturday, 3: Sunday/Holiday |
|  | *complete* | bool | True if the timeseries of the station covers more than 80 % of all days in the total timeframe of interest. |
|  | *sqv* | bool | True if the observed 2019 average norm-weekday count fits the traffic model with $SQV \geq 0.6$ |
|  | *daily_value* | int | Daily total count |
|  | *1*<br>*...*<br>*24* | int | Hourly counting values |

## Appendix C: Diurnal Cycles of all Vehicle Classes

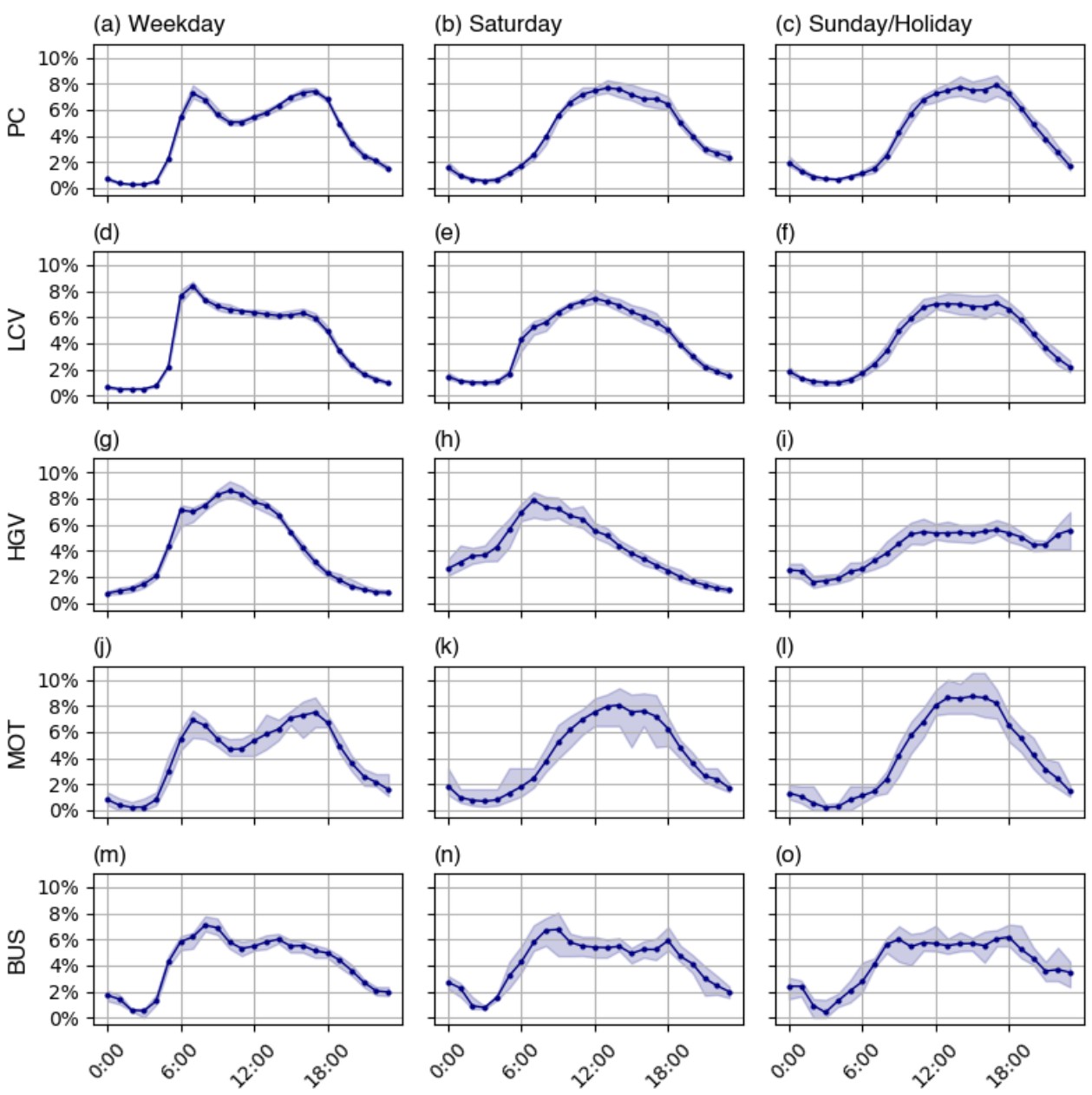

**Figure C1.** Individual cycles of all vehicle classes where emission factors are available in HBEFA. The blue line represents the average cycle for all months of the year, while the blue-shaded area indicates the range within which the cycles of individual months fall.

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
