# Peer review of "DRIVE v1.0: A data-driven framework to estimate road transport emissions and temporal profiles"

_EGUsphere, 2025_

## Author Comment (AC1)

Dear reviewer,

We highly appreciate your valuable feedback and comments, which help us significantly improve our MS. We would especially like to thank you for the good rating in the review criteria of scientific significance, scientific quality, and presentation quality. On your concerns regarding the reproducibility and traceability of the results, we would like to comment as follows.

As noted in the MS and discussed by Arioli et. al. 2020, data availability is a major challenge when developing traffic emission inventories. To a certain extent, our approach is tailored to the data availability in our target city, Munich. But these are not particularly specific data sources - they are also available in a similar form in numerous other cities. Most medium or larger-sized cities maintain macroscopic traffic models for road transport planning and analyzing travel demand scenarios. Additionally, vehicle-specific traffic counting data is also very common but is hardly available to the public as a harmonized data set and is usually restricted to municipal authorities. To the best of the author's knowledge, the largest, harmonized, multi-city counting dataset is UTD19 (Loder et al., 2019), which provides 2 years of traffic counting data for 40 cities globally. It demonstrates the global availability of such data and thus the reproducibility in other cities. Nevertheless, we pass the responsibility for obtaining the required data on to future users of the model. In our specific case, the comprehensive implementation required the use of city-specific, non-English data sources that cannot be shared publicly.

Finally, the HBEFA emission factor database is a well-established and commonly used emission factor database in Europe. It is used directly by several European countries (e.g., Germany, Switzerland, Austria, France, Sweden, Norway) for official reporting and indirectly through integration into other widely used tools such as COPERT. While HBEFA provides average fleet emission factors based on current national fleet statistics for some countries, users can also create custom fleet compositions representative of other countries. This, however, is outside the scope of the presented methodology. We utilized a comprehensive licensed version, which enables detailed differentiation between emission types (including hot exhaust and cold start), pollutants, model years, traffic conditions, and road categories. Due to licensing restrictions, this version cannot be made publicly available. Aggregated HBEFA emission factors are available as a free online version[1].

To support future users, we provide comprehensive documentation and a step-by-step user manual alongside the code repository, available on both GitHub and Zenodo. All computational notebooks specify the necessary procedures, and detailed README files guide users through each stage of implementation. The required data formats are explicitly defined, ensuring a streamlined model application. This MS is submitted to GMD as a model description paper. It includes the main article and a code repository; all provided with enough technical detail to enable reimplementation of the methods in other geographical regions.

We added the following paragraph in the conclusions to communicate the limitations and applicability more clearly.

| 502 | A limitation to the application of the framework is the availability of data in the target city. The model requires a macroscopic traffic model and vehicle-specific counting data for its calculations. However, in well-developed cities, both data sets are usually |
|---|---|
* * *
[1] https://www.hbefa.net/de/software#online-version

| | available and can be requested from the city administration if they are not publicly available. In addition, the HBEFA emission factor database is specifically tailored to European cities and is best applied in countries directly supported by the database, such as Germany, Switzerland, Austria, France, Sweden, and Norway. HBEFA emission factors are based on national vehicle fleets, driving patterns, and fuel characteristics, which may not fully represent conditions in other regions. Still, users can also create custom fleet compositions representative of other countries, although this falls outside the scope of the presented methodology. |
|---|---|

1. Reviewer comment: Moreover, the authors' approach to computing road transportation emissions is not novel. Similar methods can be found in the DARTE and Vulcan models (Gately, Hutyra and Sue Wing, 2015; Gurney et al., 2020). In their revision, I would encourage the authors to refer to previous attempts at modeling road transportation emissions and highlight how DRIVE is novel or different.

**Response:** Both DARTE and Vulcan v3.0 use distinct methods to estimate on-road traffic emissions, utilizing traffic counting data from the US HPMS (Highway Performance Monitoring System). While Vulcan v3.0 redistributes county-scale FFCO2 estimates based on Average Annual Daily Traffic (AADT) counts for higher-level roads and a road-length-based proxy for local roads, DARTE performs a bottom-up calculation based on road-link-level activity data. The methodology of DARTE is similar to that of DRIVE, and it was added to the introduction.

| 52 | DARTE provides a national on-road $CO_2$ inventory for the US based on average annual daily traffic counts (AADT) from highway counting stations |
|---|---|

Furthermore, our novelty is the comprehensive use of vehicle counting data for (1) daily temporal extrapolation of the traffic model on different road types, (2) daily estimation of the vehicle shares on different road types, (3) data-based prediction of the traffic condition at a road-link level to apply traffic-related emission factors. This enables an exact temporal profile and emission calculation and facilitates real-time capability of the method. We also show a data-based uncertainty assessment of the activity data and the emission estimate. In addition, our method is based on free, anonymous data and is applicable in countries in Europe with strict data protection regulations.
We further highlighted our novelties in the introduction.

| 67 | Our novelty is the extensive use of counting data from more than 100 individual traffic detectors across different road types within our city of interest. First, the local counting data is used to extrapolate the traffic model over time to accurately estimate the daily traffic volume on each road segment. Next, we introduce a novel method for data-based and time-resolved calculation of the vehicle share. The vehicle class-specific temporal scaling of traffic activity enables a precise and data-based prediction of traffic conditions to apply traffic-related emission factors. |
|---|---|

2. Reviewer comment: The authors calculate emissions for Munich, Germany, using a lot of European and German data. However, they do not state whether the model they present is limited to Germany or Europe.

**Response:** All methods for processing traffic activity data, including temporal extrapolation, splitting of vehicle shares, and estimation of traffic conditions, can be applied wherever the necessary traffic data is available. Although data availability is a significant limitation, similar traffic models and counting data are prevalent worldwide. The HBEFA emission factor database is specifically tailored to European countries by providing average fleet

compositions or average traffic situations based on national statistics. As stated in the introductory comment, users can also create custom fleet compositions representative of other countries. This, however, is outside the scope of the presented methodology. We added the following to the conclusion to communicate the limitations more clearly.

| 502 | A limitation to the application of the framework is the availability of data in the target city. The model requires a macroscopic traffic model and vehicle-specific counting data for its calculations. However, in well-developed cities, both data sets are usually available and can be requested from the city administration if they are not publicly available. In addition, the HBEFA emission factor database is specifically tailored to European cities and is best applied in countries directly supported by the database, such as Germany, Switzerland, Austria, France, Sweden, and Norway. HBEFA emission factors are based on national vehicle fleets, driving patterns, and fuel characteristics, which may not fully represent conditions in other regions. Still, users can also create custom fleet compositions representative of other countries, although this falls outside the scope of the presented methodology. |
|---|---|

 Reviewer comment: The authors refer to HBEFA emission factors in multiple parts of the paper, but there is no citation or reference to a table in the paper.

**Response:** HBEFA is a licensed product (standard price: 250€, student version: 50€[2]) and is provided as an MS-Access database, which cannot be shared publicly. References to the latest documentation of the database were added to the introduction.

| 41 | The activity data can be combined with a range of emission models such as HBEFA (Handbook Emission Factors for Road Transport) (Notter et al., 2019, 2022). |
|---|---|

4. Reviewer comment: The authors use a macroscopic traffic demand model maintained by the City of Munich. Unfortunately, this model is not cited, and the source is not mentioned. It is also unclear whether the model is open source or publicly/freely available.

**Response:** The traffic model used in this study is not publicly available. It was provided by the city free of charge, along with the model documentation, after signing a data transfer agreement. Yet it is also not necessary for the model to be publicly available; it must exist for the targeted city, as it does for most cities. We clarified the model origin in the MS.

| 101 | The model is not publicly available but was provided free of charge by the city administration after signing a data transfer agreement. Generally, it is |
|---|---|

5. Reviewer comment: The authors use traffic counting data from the city administration and BASt. Citations and sources are missing. From the description provided by the authors, it appears that the data sources might be in German. If this is true, it would be an additional hindrance to reproducibility.

**Response:** Yes, both data sources are in German. Like the macroscopic traffic model, the traffic counting data from the city administration was provided free of charge after signing a data transfer agreement. It is city-specific and cannot be shared publicly. The BASt counting data is publicly available under the CC BY 4.0 license. We clarified the origin of both data sources in the MS.

| 117 | Like the traffic model, the traffic counting data from the city administration is not publicly available but was provided free of charge after signing a data transfer |
|---|---|
* * *
[2] https://www.hbefa.net/en/order-form

| | |
|---|---|
| | agreement. Counting data from the BASt is shared publicly under the CC BY 4.0 license. |

6. Reviewer comment: It was not clear how the correction factor $k_{i,vc}$ was calculated.

**Response:** We revised section 2.2.2 for more clarity.

| | |
|---|---|
| 186 | We apply additional correction factors because vehicle shares vary spatially, even on roads of the same type. For example, ring motorways have a higher HGV share than radial motorways (routes into the city center). The traffic model provides vehicle class-specific traffic volumes for HGV and LCV. They were used to infer average annual weekday shares $\delta_{i,HGV}^{model}$ and $\delta_{i,LCV}^{model}$ for each road link. Subsequently, we calculate the average weekday share of the reference year 2019 based on the counting data $\delta_{HGV,r}^{count,ref}$ and $\delta_{LCV,r}^{count,ref}$ for each road type. The quotients $\kappa_{i,HGV}$ and $\kappa_{i,LCV}$ between the vehicle share in the traffic model and at the counting stations, aggregated by road category, were used to correct the vehicle share at each road link. The correction factor $\kappa_{i,vc}$ for the remaining vehicle categories is calculated by dividing the total share left after correcting for HGV and LCV by the original, uncorrected share for these categories. This factor ensures that the sum of all shares $\delta_{vc,r}$ equals one after correction. |

7. Reviewer comment: Table B2 (Appendix B) describes scaling factors for Passenger Car Units. However, a description of how or why such scaling factors were used is missing.

**Response:** According to another reviewer's comment, we split the table and moved the passenger car equivalent scaling factors to the related section in the MS and revised the table caption for more clarity.

| | |
|---|---|
| 209 | The traffic volume for each vehicle class is converted into passenger car equivalents (PCE) to account for differences in vehicle size and impact on traffic flow. This conversion enables the analysis of mixed traffic streams as if they consisted solely of passenger cars. The scaling factors $\eta_{vc}$ applied are shown in Table 3. |

**Table 3.** Passenger Car Equivalent (PCE) scaling factors $n_{vc}$. These factors are applied to adjust the mixed traffic stream for the size and flow impact of different vehicle categories.

| vehicle class $vc$ | PCE factor $n_{vc}$ |
|---|---|
| PC | 1 |
| MOT | 1 |
| LCV | 1 |
| HGV | 2.5 |
| BUS | 1.75 |

8. Reviewer comment: All German citations lead to reproducibility issues.

**Response:** We appreciate the reviewer's emphasis on reproducibility but respectfully disagree with the general statement that all German citations lead to reproducibility issues. We believe the situation is similar to all non-Englisch speaking countries. In the end, the users of our inventories are also the local governments/city authorities. The framework we propose depends on local data sources, which are often only fragmentarily available. In addition, the assessment of the model results was based on local information (e.g., the emissions

assessment of the city of Munich). The German sources cited were used as a reference to local data sources, to evaluate national emission factors, and to compare the model results with local or national results. All key data, methods, and analyses are described in detail to ensure transparency and reproducibility. We think the situation is reproducible for many cities in the world.

9.  Reviewer comment: In section 3.2, a citation for 'emission regulation' is missing.

**Response:** We reference the German National Inventory Report (IIR), which shows a similar trend and draws the same conclusion.

| 282 | This change can be attributed to the ongoing development of emissions legislation, which improves the emissions performance of the entire fleet (Gniffke et al. (2024). |

10. Reviewer comment: The authors have often used qualitative terms like "a substantial portion". I would encourage them to quantify these.

**Response:** We revised the MS at multiple locations.

| 17 | Approximately 20-50 % of urban greenhouse gases and air pollutants are associated with transportation, mainly from road vehicles (Chapman, 2007; Edenhofer et al., 2014; Crippa et al., 2021). |
| 31 | A sub-kilometer spatial resolution and hourly temporal resolution become essential when inventories are used alongside observations and atmospheric transport models in cities. |
| 310 | Based on local activity data, the DRIVE inventory assigns 90 % of the total $CO_2$ emissions to the main road network (motorway, primary, and secondary roads), reflecting traffic volumes and congestion. |

11. Reviewer comment: DRIVE is built for Munich, a city in Germany. The authors do not explain why they focused on this city and did not develop a model for Europe or even a global one.

**Response:** Clarifications were added to the MS.

| 77 | The framework was developed and implemented for the City of Munich as part of the ICOS Cities project. ICOS Cities aims to develop and evaluate standardized greenhouse gas measurement and services in urban environments, with Munich as one pilot city. |

12. Reviewer comment: Related to the above point, the authors do not elaborate on whether this model can be extrapolated to the rest of Europe or the globe.

**Response** (identical to response on comment 2): All methods for processing traffic activity data, including temporal extrapolation, splitting of vehicle shares, and estimation of traffic conditions, can be applied wherever the necessary traffic data is available. Although data availability is a limitation, similar traffic models and counting data are prevalent worldwide. The HBEFA emission factor database is specifically tailored to European countries by providing average fleet compositions or average traffic situations based on national statistics. As stated in the introductory comment, users can also create custom fleet compositions representative of other countries. This, however, is outside the scope of the presented methodology. We added the following to the conclusion to communicate the limitations more clearly.

| 502 | A limitation to the application of the framework is the availability of data in the target city. The model requires a macroscopic traffic model and vehicle-specific counting data for its calculations. However, in well-developed cities, both data sets are usually available and can be requested from the city administration if they are not publicly available. In addition, the HBEFA emission factor database is specifically tailored to European cities and is best applied in countries directly supported by the database, such as Germany, Switzerland, Austria, France, Sweden, and Norway. HBEFA emission factors are based on national vehicle fleets, driving patterns, and fuel characteristics, which may not fully represent conditions in other regions. Still, users can also create custom fleet compositions representative of other countries, which falls outside the scope of the presented methodology. |
|-----|---|

13. Reviewer comment: The HBEFA emission factors are national-scale factors, but DRIVE is for a specific city. The authors need to address the variation of factors within a country and/or whether the national-level statistics are representative of the city of Munich.

**Response:** The base emission factors originate from a national-level database; their parameterization and application were localized to reflect Munich's conditions. HBEFA provides distinct fleet compositions for different area- and road-types. We select emission factors for the area type "urban" and consider the different road types available in the target region.
We fully acknowledge that uncertainty remains due to regional variability and briefly discussed it in the limitations of our uncertainty assessment and the conclusion.

| 442 | Furthermore, on a city level, no specific information is available regarding the fleet composition, such as powertrain technologies and emission concepts, so statistical averages provided in HBEFA are used. These factors can vary significantly based on vehicle type, age, maintenance, and operating conditions, which may not be fully represented in a generalized dataset. |
|-----|---|
| 507 | Despite the limitations in accurately modeling traffic conditions and the limited knowledge of the local fleet composition, the proposed method provides a comprehensive, data-driven, and scalable approach to exploit static travel demand models and counting data from multiple traffic counting stations to estimate road transport emissions and their uncertainty. |

14. Reviewer comment: While introducing the macroscopic traffic demand model for Munich, the authors do not describe its spatial resolution.

**Response:** Clarifications were added to the MS.

| 102 | The road network is split into road segments, which are represented as lines in the model. The spatial resolution ranges from several tens of meters in densely networked inner-city areas to a kilometer scale on highways. |
|-----|---|

15. Reviewer comment: In section 2.1.3, the authors need to elaborate on what "manual data curation" and "automatic preprocessing" mean and/or involve.

**Response:** Clarifications were added to the MS. The required steps for data curation and preprocessing are presented in section 2.1.3. In this chapter, we provide a broad overview while offering detailed comments on our implementation in the related computational notebooks and supplements.

| 130 | The exact steps required depend on the format and quality of the available data and may differ in other cities. The basic procedure is described in the following section. |
|---|---|

16. Reviewer comment: Section 2 of the manuscript is the 'Methodology' section. However, the authors include some results in this section. These should be moved to the proper section.

**Response:** We appreciate the observation regarding the inclusion of some results in the methodology section. However, we would like to clarify that these are not intended as final results but as intermediate outcomes to help the reader better understand the method presented. We show the temporal scaling factors applied to extrapolate the traffic model, which includes annual cycles, diurnal cycles, and the applied vehicle shares. We prefer to maintain the current structure, as it enhances comprehension for the readership.

17. Reviewer comment: The universal and/or SI system uses a dot (".") as a decimal point and a comma (",") as a separator. The authors should use these standards consistently in the manuscript and not use the German system.

**Response:** Thank you for this comment. We have incorporated the suggested changes.

18. Reviewer comment: Section 3.1 starts talking about 'LOS' without introducing or explaining the acronym previously.

**Response:** We clarified the acronym in section 2.3.

| 203 | HBEFA distinguishes 365 different traffic situations by considering the road type, road gradient, speed limit, area type (rural vs. urban), and the level-of-service (LOS) (Notter et. al., 2019). The road type, gradient, and speed limit are static for each road link, and the area type "urban" is used for the whole city area. The LOS reflects the prevailing traffic condition and is estimated for each road link using the volume-capacity ratio $x_i$ (equation 2) and dedicated thresholds to distinguish between five classes: Freeflow, Heavy, Saturated, Stop&Go, and Stop&Go2 (gridlock with average speeds of 5-10 km/h). |
|---|---|

19. Reviewer comment: In the same section, the authors also do not explain 'Stop & Go' and 'Stop & Go 2'.

**Response**: We clarified the traffic conditions in section 3.1.

| 258 | In the HBEFA, the LOS class Stop&Go describes congestion with frequent stops and slow traffic, while Stop&Go2 stands for heavy congestion or traffic jams characterized by average speeds of 5-10 km/h. |
|---|---|

20. Reviewer comment: While describing 'well-to-tank' and 'tank-to-wheel' emissions, the authors need to explain how they avoid double-counting (e.g., oil tanker trucks can feature in both).

**Response:** We thank you for pointing out the issue! It should be Well-to-wheel (WTW) emissions in this case, which is the sum of well-to-tank and tank-to-wheel. Consequently, double counting does not need to be discussed in this context. We changed accordingly.

21. Reviewer comment: To scale weekday traffic volume to the weekend, the authors multiply by 0.8. The source (or citation) of this number needs to be explained.

**Response:** We use the traffic counting data to scale weekday traffic volume to the weekend. The factor of 0.8 is used in the method employed by the responsible department (RKU) in the city administration to account for reduced traffic on weekend days. The city's calculation method is not published, and the factor was shared during an online meeting. Therefore, a referenceable source is not available.

22. Reviewer comment: Figure 6: Different vehicular types have very different magnitudes, so I recommend using different color palettes for each map to demonstrate distinct spatial patterns better. Magnitudes can be contrasted using a simple bar plot or a pie chart.

**Response:** We appreciate the reviewer's suggestion. A new figure and caption have been created.

[Figure]

*Figure 6. Spatial distribution and total share of $CO_{2,ff+bf}$ emissions of different vehicle classes in 2019. Passenger cars (PC) account for the largest share in emissions (67%). Followed by heavy goods vehicles (16%) and light commercial vehicles (15%). High emission values on the main roads are visible for all vehicle classes. While the emissions from HGV concentrate on main roads and the motorway, other vehicle classes emit on all roads. No distinct spatial patterns can be observed for BUS and MOT due to the absence of spatial information in the traffic model. In total, buses and motorcycles only constitute about 3% of Munich's emissions.*

23. Reviewer comment: Figure 7: The top row figures lack a color bar, and the bottom row figures need a quantitative color bar rather than a qualitative one. Moreover, I recommend plotting a percentage relative difference map to show spatial patterns of differences.

**Response:** Again, we appreciate the reviewer's suggestions and updated the figure and related caption.

304

[Figure]

*Figure 7. Comparison of the spatial distribution of $CO_{2,ff+bf}$ for the traffic sector from three inventory datasets. Plots (a)-(c) are normalized to the respective total values of the city to present them on a uniform scale. Therefore, each cell value represents the fraction of the total emissions attributed to this cell. TNO (a) attributes major emissions to the city center, i.e., the place with the highest population density. The UBA shows a more homogeneous spatial distribution, and compared to the DRIVE inventory, roads with high traffic volumes are less pronounced. The difference plots in (d) and (e) show the absolute difference between the normalized cell values. They indicate that UBA and TNO attribute lower emissions (blue) to parts of the main road network and higher emissions (red) to minor roads. DRIVE uses validated local traffic activity data, more accurately representing the spatial distribution of related traffic emissions. Both downscaled inventories reflect the incorporated spatial proxies.*

**Sources**

Arioli, M. S., D'Agosto, M. D. A., Amaral, F. G., and Cybis, H. B. B.: The Evolution of City-Scale GHG Emissions Inventory Methods: A Systematic Review, Environmental Impact Assessment Review, 80, 106 316, https://doi.org/10.1016/j.eiar.2019.106316, 2020.

Notter, B., Keller, M., Althaus, H.-J., Cox, B., Knörr, W., Heidt, C., Biemann, K., Räder, D., and Jamet, M.: HBEFA 4.1 Development Report, Development report, INFRAS, Sennweg 2, 3012 Bern, 2019.

Notter, B., Cox, B., Hausberger, S., Matzer, C., Weller, K., Dippold, M., Politschnig, N., Lipp, S., Allekotte, M., Knörr, W., Andre, M., Gangnepain, L., Hult, C., and Jerskjö, M.: HEBFA 4.2 Documentation of Updates, Update report, INFRAS, Sennweg 2, 3012 Bern, 2022.

Loder, A., Ambühl, L., Menendez, M. *et al.* Understanding traffic capacity of urban networks. *Sci Rep* **9**, 16283 (2019). https://doi.org/10.1038/s41598-019-51539-5

Gately, C. K., Hutyra, L. R., & Sue Wing, I. (2015). Cities, traffic, and CO2: A multidecadal assessment of trends, drivers, and scaling relationships. *Proceedings of the National Academy of Sciences*, *112*(16), 4999–5004. https://doi.org/10.1073/pnas.1421723112

Gurney, K. R., Liang, J., Patarasuk, R., Song, Y., Huang, J., & Roest, G. (2020). The Vulcan Version 3.0 High-Resolution Fossil Fuel CO 2 Emissions for the United States. *Journal of Geophysical Research: Atmospheres*, *125*(19). https://doi.org/10.1029/2020JD032974

---

## Author Comment (AC2)

Dear reviewer,

We highly appreciate your valuable feedback and comments, which help us significantly improve our MS. We would like to thank you very much for your detailed evaluation, in which you positively acknowledge our workflow, the data source description, and the political relevance. We particularly welcome your methodological suggestions for expanding the diagnostics, quantifying potential biases, and clarifying assumptions related to the presented emission estimates. Please find our response to your comments below.

1. Reviewer comment: In Section 2.3 (p. 9), LOS thresholds are tuned so that VKT shares match national FCD data within 1%. This calibration is critical for emission factor assignment, yet the actual threshold values per road class and the pre-/post-VKT distribution are not presented. Please report these in the main text. In addition, discuss whether using national FCD distributions for an urban network dominated by signalized intersections introduces systematic bias, and quantify the sensitivity of NOx and CO to a ±10% shift in all thresholds.

**Response:** We moved Table S2 from the supplements to the main text, which shows the calibrated VCR thresholds and the nominal thresholds derived from HBS, 2015, which served as the starting point. The resulting distribution of the total VKT to different traffic conditions (LOS classes) is presented in Figure 5b for all years. Additionally, we introduce a new section, "Sensitivity to Specific Model Parameters", which covers the proposed sensitivity analysis to a ±10% shift in all thresholds.
Finally, we would like to note that distinct national distributions for rural and urban areas are available. We selected the distribution for urban areas, which was further clarified in the MS. While the volume-capacity ratio is just a simple proxy to estimate traffic conditions, we assume no systematic errors on a city level, thanks to the applied optimization of the VCR thresholds. This is briefly discussed in section 5.4 Limitations of the Uncertainty Assessment.

| 214 | Schmaus et al. (2023) investigated the distribution of the vehicle kilometers traveled (VKT) among different LOS classes in Germany based on floating car data. He shows distinct distributions for rural areas and agglomerations, whereby we employed the distribution for agglomerations in this study. Although this corresponds to a national average for urban areas, we assume that it reflects the situation in Munich well. |
|---|---|
| 364 | 4 Sensitivity to Specific Model Parameters
The approach is subject to fine-tuning of parameters and some heuristic assumptions that can affect the final emissions result. In the following section, we will examine the sensitivity of the estimate to changes in the VCR thresholds and the cold start allocation radius.

4.1 Sensitivity Analysis of VCR Thresholds
In section 2.3, we propose to optimize the VCR thresholds to match the national distribution of traffic situations on each road type within ±1%. This step is crucial for selecting the correct emission factor, as emissions increase sharply and non-linearly in congested traffic conditions. We tested a scenario with ±10% for all thresholds after the optimization, which is referred to as the nominal scenario. Table 6 shows the result of these scenarios and indicates that the emissions increase by 7 to 10 % if all VCR thresholds are lowered by 10 %, and the emissions decrease by 3 to 5 % if the thresholds are increased by 10 %. Raising the thresholds will lead to an increase in free-flow conditions by 7 %, and lower Stop&Go conditions by 4 %. |

Lowering results in a 7% decrease of VKT under free-flow conditions and a 6% increase of Stop&Go conditions. We conclude that a change in the thresholds and the associated distribution of VKT across the traffic situations has a severe impact on the resulting emissions, and the optimization must be conducted with great care. The distribution of VKT based on city-specific statistics or the allocation of traffic conditions based on floating car data would further increase local representativeness.

**Table 8.** Emission and VKT-distribution sensitivity to a ±10% change of all VCR thresholds. The nominal scenario corresponds to the optimized threshold values used for the emissions calculation and shown in Table 4.

| | Emissions | | | Traffic Situations | | | | |
|---|---|---|---|---|---|---|---|---|
| | $CO_2$ [kt] | $CO$ [t] | $NO_x$ [t] | Freeflow | Heavy | Satur. | St&Go | St&Go2 |
| **Nominal scenario** | 1287 | 2583 | 3333 | 53.4% | 22.5% | 16.3% | 5.6% | 2.2% |
| **Thresholds -10%** | 1403 | 2760 | 3664 | 46.6% | 21.7% | 17.6% | 8.2% | 6.0% |
| *rel. change* | *+ 9.0%* | *+ 6,9%* | *+ 9,9%* | *- 6.8%* | *- 0.9%* | *+ 1.3%* | *+ 2.7%* | *+ 3.7%* |
| **Thresholds +10%** | 1226 | 2519 | 3153 | 60.1% | 22.0% | 14.2% | 2.7% | 1.0% |
| *rel. change* | *- 4.7%* | *- 2.5%* | *- 5.4%* | *+ 6.7%* | *- 0.5%* | *- 2.1%* | *- 2.9%* | *- 1.2%* |

2. Reviewer comment: Section 2.5 fixes a 1.5 km allocation radius based on an assumed travel time at 60 km/h. This assumption may not hold across all road types and congestion states. Please provide a sensitivity analysis (e.g., 0.8 km, 2.0 km) to show the impact on the spatial allocation of cold-start emissions. Also, all temperature binning uses a single urban station. Given the size of the domain, is this representative? Finally, Figure 8 and p. 16 note negative NOx cold-start factors above 25°C. Clarify whether such negative factors can lead to negative hourly or link-level totals and whether you impose a non-negativity constraint.

**Response:** We further analyse the sensitivity to the applied allocation radius in a new section, "Sensitivity to Specific Model Parameters".
We briefly discuss the limited temperature representativeness in Section 2.5.
Finally, we clarify how cold-start surcharges can be negative above 25°C in the related figure caption.

| 235 | This measured temperature is not fully representative of every vehicle start in the study area, but it does provide a practical, time-resolved reference value in Munich for the application of the emission factors. Further influences, such as the parking location of the vehicle (e.g., underground garage, carport, street parking), cannot be examined in detail. |
|---|---|
| 348 | *(figure caption)*
Negative cold start surcharges for NOx and NO2 are plausible, as these only represent a surcharge to the hot emissions. This means that in this case, the emissions during cold start are lower than the hot emissions. Overall, there are no negative emissions. |
| 364 | 4.2 Sensitivity Analysis of Cold-Start Allocation Radius
The number of vehicle starts is distributed across spatial zones in the traffic model and available for PC and LCV. We assign vehicle starts to all intersecting road links within the zone and a surrounding 1.5 km buffer radius, weighted by the traffic volume of the respective road link. Motorways and primary roads are generally excluded. To test the influence of the allocation radius, two additional scenarios with a 0.8 km and a 2 km buffer radius were tested. A study by Pina and Tchepel (2023) shows a typical driving |

distance of 5 km for inner-city journeys under cold start conditions, with excess emissions being highest at the start of the journey and then decreasing exponentially. We conclude that changing the allocation radius does not change the total emission at a policy-relevant level. Lowering the buffer radius generally leads to an increase in cold starts, attributed to residential roads as shown in Table 9. Figure 9 shows a difference map between allocated cold start emissions of the nominal scenario and 800 m and 2 km buffer radius, respectively. Larger differences are visible outside the city center, particularly for 800 m scenario. However, neither map shows a systematic correlation between the buffer radius and spatial distribution that could indicate an inadequate assumption, and the buffer distance has little influence on the city's total. The 1.5 km radius is applied until more conclusive information becomes available.

**Table 9.** Sensitivity of results when changing the cold start buffer radius. Total emissions change by less than 1 %.

| | Total Emissions | | | Road Types [starts/day] | |
|---|---|---|---|---|---|
| | $CO_2$ [kt] | NOx [t] | CO [t] | Secondary | Residential |
| **Nominal scenario** | 22.46 | 64.16 | 1549.23 | 2127309 | 806793 |
| **Buffer = 0.8 km** | 22.33 | 63.77 | 153975 | 2106709 | 827393 |
| *rel. change* | - 0.6 % | - 0.6 % | - 0.6 % | - 1.0 % | + 2.6 % |
| **Buffer = 2 km** | 22.67 | 64.75 | 1563.39 | 2137269 | 796834 |
| *rel. change* | + 0.9 % | + 0.9 % | + 0.9 % | + 0.5 % | - 1.2 % |

[Figure]

**Figure 9.** Relative difference in total cold start emission surcharges with an allocation buffer radius of a) 2 km and b) 800 m on a 1x1 km grid cell level. Larger relative differences can be observed outside the city center. However, no further systematic correlation between buffer radius and spatial distribution can be observed.

3. Reviewer comment: The discussion on p. 15 attributes much of the CO difference with UBA and TNO to the assumed uniform 120 km/h motorway limit. While I understand that counter-based speed data have limitations, "unreliable" (Section 2.1.2) is too vague - please quantify coverage or bias. Even partial speed information could help construct a more realistic speed distribution. Consider adding a scenario with higher or unrestricted motorway speeds to estimate the impact on CO and report CO contributions by road class.

**Response:** Unfortunately, counter-based speed information is not available for any motorway section but only for inner-city stations (cp. Figure 1). We clarified this in section 2.1.2. The maximum speed limit of 120 km/h on the motorway is a model parameter of the macroscopic traffic model that we received from the city's mobility department. However, the speed limit on the motorway is variable and is controlled according to traffic volume, weather conditions, existing roadworks, or accidents.
To further enrich our discussion in section 3.3, we show the contribution of CO from the motorway to the total hot exhaust emissions in 2019 and estimate an additional scenario with the aggregated EF for the road category "Motorway".

| 121 | Some stations also provide the average speed of vehicles, but this data is not used in the model as it was deemed unreliable. We observed numerous artifacts and outliers in the speed data, which we attributed to stop-and-go traffic, intersection effects, and maintenance issues. Moreover, there is no speed information available for the motorway (BASt counters), which makes it impractical to use this data consistently for all major road types. |
|-----|-----|
| 327 | The motorway accounts for approximately one-third of the total VKT in this study, and 40% of the total hot exhaust emissions (total hot CO = 2583 tons; motorway hot CO = 1032 tons). Motorway-type road links in the traffic model used have a maximum speed limit of 120 km/h. In reality, however, the allowed speed is regulated depending on the traffic load, and on German motorways at free-flow conditions, no speed limit is applied. To further evaluate the impact of high free-flow speeds on the motorway, we applied the national, aggregated emission factor for motorways to all motorway road links in our study. This triples the CO contribution from the motorway (motorway hot CO = 3539 tons), resulting in a total CO emission of 6701 t and a $CO_{2,ff+bf}$/CO ratio 195.8. This suggests that we probably underestimate CO emissions on the highway, while methods based on proxies overestimate the urban share, where motorway speeds are generally lower due to high loads. |

4. Reviewer comment: The mapping from 8+1 counter classes to HBEFA categories is in Appendix B2 but is central to your method. This should be moved into the main paper or SI. The spatial correction factor κ is derived from weekday averages; please comment on whether this remains valid for weekends/holidays. If possible, validate κ-corrected class shares at the 64 independent stations, not only total volumes.

**Response:** According to your suggestion, we split Table B2 and moved the vehicle category characterization to Section 2.1.3, and the PCU scaling factors to Section 2.3.
To test the spatial correction on weekend days, we validated the κ-corrected modelled daily vehicle-specific traffic volumes against daily traffic counts for 2019. We colorize different road types and sub-select weekdays and weekend days. The result was added and discussed in the supplement Section 4. From the analysis, we conclude that the spatial correction is valid on weekend days.

| 138 | **Table 1.** Aggregation of vehicle categories from 8+1 counting data categorization to HBEFA compatible vehicle classes. |
|-----|-----|

| 8+1 vehicle class | HBEFA vehicle class ($vc$) |
|---|---|
| Passenger Car | PC - Passenger Car |
| Passenger Car w. Trailer | |
| Motorcycles | MOT - Motorcycles |
| Light Truck | LCV - Light Commercial Vehicle |
| Truck | |
| Truck w. Trailer | HGV - Heavy Goods Vehicles |
| Truck w. Semi-Trailer | |
| Bus | BUS - Coach |
| Not Classified | - |

| 212 | **Table 3.** Passenger Car Equivalent (PCE) scaling factors $n_{vc}$. These factors are applied to adjust the mixed traffic stream for the size and flow impact of different vehicle categories. |
|---|---|

| vehicle class $vc$ | PCE factor $n_{vc}$ |
|---|---|
| PC | 1 |
| MOT | 1 |
| LCV | 1 |
| HGV | 2.5 |
| BUS | 1.75 |

| SI 53 | Upon request, the city department responsible argued that no LCV-specific calibration had been performed. Misclassifications between PC and LCV of the double loop traffic detectors could also play an important role. The fit is particularly worse at trunk roads where only double loop detectors are installed, compared to the motorway, where a more robust camera-based classification is in place. We are aware of the data-model mismatch, but we have more confidence in the traffic model. |
|---|---|

| SI 69 | 4.1 Validation of modeled, vehicle-specific and κ-corrected daily traffic volumes
To model the traffic volume on each road link, we apply road and vehicle class-specific temporal extrapolation to the average weekday traffic volume provided by the traffic model. In addition, vehicle-share correction factors κ were applied to account for different modal splits on the same road type (example: higher HGV share on the ring motorway vs. on radial motorways into the city). In Figure S4 and S5, κ-corrected, modeled vehicle count and daily counts from counting stations on the same road link are shown. Figure S4 shows the daily traffic volume for all weekdays in 2019. A good fit for SUM, PC, and HGV can be observed across all road types. LCV, MOT, and BUS have worse fit statistics and higher variance. This is related to the small daily counts of MOT and BUS and the already worse fit of LCV in the traffic model (cp. Figure S2). The modelled traffic volume underestimates MOT and BUS and overestimates LCV. Additionally, we observe no significant difference but only slightly lower R2-values for weekend days compared to weekdays. In particular for SUM, PC, and HGV, we conclude that κ-corrected daily traffic volumes are equally valid on weekdays and weekends. |
|---|---|

[Figure]

**Figure S5.** Comparison of $\kappa$-corrected modeled vs. counted daily traffic volume on weekdays (Monday-Friday). The daily modeled count for SUM, PC, and HGV shows a good fit and a high $R^2$. For MOT, bus, and LCV, the fit statistics are less satisfactory, which is attributed to lower counts and a poorer overall fit with the traffic model.

[Figure]

**Figure S6.** Comparison of $\kappa$-corrected modeled vs. counted daily traffic volume on weekend days (Saturday, Sunday). Compared to the weekday fit, we observe a similarly high agreement for SUM, PC, and HGV, and similar deviations for LCV, MOT, and BUS. We conclude that $\kappa$-corrected vehicle volumes are as valid on weekends as they are on weekdays.

5. Reviewer comment: Equation 6 combines activity error with emission factor uncertainty, assuming independence, yet EF depends on LOS, which is derived from activity. Please test a scenario with positively correlated perturbations (for example, correlation coefficient 0.3–0.5) to illustrate the possible underestimation of total uncertainty.

**Response:** Thanks for your suggestion. Neglecting the correlation between activity and emission factor selection is clearly a shortcoming of the presented analysis. The EF selection is based on the AD, but EFs are not a continuous variable. We use discrete volume-capacity thresholds to select traffic conditions and related EFs. The relationship between AD and EF is non-linear and cannot be determined analytically. A future plan for the method is to implement a Monte Carlo simulation or use floating car data or other speed measurements to estimate the traffic condition with an independent data source. We did not implement a test with positively correlated perturbations as, in our opinion, this would not lead to a conclusive result. We would like to draw the reviewer's attention to our section "5.4 Limitations of the Uncertainty Assessment", where these and further issues are addressed.

| 438 | 5.4 Limitations of the Uncertainty Assessment |
|---|---|
| | The uncertainty analysis focuses only on hot vehicle exhaust emissions and does not consider cold start emissions due to the lack of comparable data. This approach is adequate for $CO_2$ and $NO_x$, as these emissions are mainly generated when the engine is hot. For CO, strongly influenced by excess emissions during cold starts, the analysis likely underestimates the uncertainty because cold start emissions are more uncertain than hot emissions. Furthermore, on a city level, no specific information is available regarding the fleet composition, such as powertrain technologies and emission concepts, so statistical averages provided in HBEFA are used. These factors can vary significantly based on vehicle type, age, maintenance, and operating conditions, which may not be fully represented in a generalized dataset. Moreover, estimating traffic conditions using the volume capacity ratio is a simple, robust, and scalable method, yet it is not very accurate in urban road networks. The traffic flow is more often limited by the capacity of intersections than by the road links between them. The optimization applied (section 2.3) allows us to achieve a representative distribution of traffic conditions for the whole city on an annual average. However, we cannot explicitly account for congestion effects such as queues and spillbacks. Despite these limitations, we assume the volume capacity ratio provides a reasonably accurate estimate of traffic conditions on the road link. But, at a road link level, congestion may introduce more uncertainty than reported. Finally, we do not explicitly take the correlation between traffic activity and the emission factor into account. If the traffic activity is estimated inaccurately, it leads to an incorrect traffic condition, and subsequently, a wrong emission factor is applied. A sensitivity analysis quantifying the impact of this correlation could further clarify its influence. The level of uncertainty also exhibits a daily pattern: at night, when traffic activity is low, the likelihood of a traffic jam is also low. However, during the day, especially during peak hours, the chances of experiencing traffic jams increase significantly. In future work, conducting a Monte Carlo simulation that incorporates the uncertainties related to traffic activity and emission factors during specific time periods could enable a probabilistic representation of how uncertainties propagate and better quantify the uncertainty of the emissions estimate. |

6. Reviewer comment: Section 2.2 uses the same temporal factor for all minor roads due to lack of counters. This is a practical assumption but potentially introduces bias. Please provide an upper bound estimate of the VKT/NOx error this could cause. Also report the proportion of days filled by imputation and its effect on annual totals.

**Response:** While higher-level road types are well equipped with traffic stations, this information is lacking at lower-level roads. Therefore, we aggregate the counting data for temporal scaling as mentioned in Section 2.2. Initial exploratory analysis showed that diurnal profiles are similar for different road types but differ for vehicle classes. For annual profiles, again, different vehicle classes showed more distinct features than road types. We keep individual temporal scaling profiles for different vehicle classes and aggregate them to scaling road types as shown in

Table 1.
The total NOx share for the road type *Trunk Road/Primary-National* is 0.01% and 10.65% for the road type *Access-residential.* In our assessment, scaling the road using the proposed aggregated road types does not lead to any considerable distortion, and if it does, this would only affect a small share of emissions.

*Table 1: Aggregation of traffic counting stations from different road types to three distinct scaling road types.*

| Road Type | # Counters | Scaling Road Type | # Counters |
|---|---|---|---|
| Motorway-National | 65 | Motorway | 65 |
| Trunk Road/Primary-City | 17 | Primary-City | 17 |
| Trunk Road/Primary-National | 3 | Distributor/Secondary | 64 |
| Distributor/Secondary | 60 | | |
| Access-residential | 1 | | |

The proportion of days filled differs by road type, vehicle class, and year. For the timeframe of the study (2019 until 2022; 1461 days), we imputed between 0 and 88 (6.02%) days. Table 2 provides an overview. We do not expect an effect on the annual total by this imputation.

*Table 2: Share of days filled for the timeframe of 2019 until 2022.*

| | BUS | HGV | LCV | MOT | PC | SUM |
|---|---|---|---|---|---|---|
| **Distributor/Secondary** | 0 % | 0.07 % | 0.07 % | 0 % | 0 % | 0 % |
| **Motorway-National** | 0 % | 0.41 % | 0 % | 0 % | 0.21 % | 0.27 % |
| **TrunkRoad/Primary-City** | 2.46 % | 3.63 % | 3.97 % | 2.46 % | 5.54 % | 6.02 % |

7. Reviewer comment: Figure 9 shows systematic overestimation at high volumes. Please include a breakdown of errors by road class and LOS to help identify whether this bias is linked to particular conditions. Also, specify how many stations were excluded from validation and their spatial distribution.

**Response:**
While re-running the notebook to investigate the differences, we found an error when importing the counting data into the notebook. Before centrally defining all data paths in *data_paths.py,* we explicitly imported data at the beginning of the notebook. The import filename was not changed in the related file where we calculated the activity and emission uncertainty, which led to importing old counting data (from 27.02.2024). In this old counting dataset, we collectively excluded all counting stations that did not provide data for all vehicle classes, leading to the exclusion of many counting stations along the motorway that only

provide total traffic volume counts in 2019. We actualized Figures 10 (previously Figure 9) and 11 (previously Figure 10) in the MS and Figure S2 in the supplements.

We assigned valid counting data to 82 road links within the city boundaries. The valid flag requires an SQV > 0.6 and data availability for 2019. A map of valid and ivalid flagged counting stations was added to the supplements (Figure S4)

Furthermore, we updated Figure 10 (previously Figure 9) to include a breakdown into different road classes and discuss it in the MS.

A breakdown by LOS is generally not possible for daily and annual aggregated figures, as the traffic situation varies throughout the day and the year. In addition, the traffic situation is a function of traffic volume and varies depending on whether actual count data or modelled traffic volume is used as the basis for the calculation. Both may result in a different volume-capacity ratio and thus lead to different traffic situations. For these reasons, we did not implement the classification based on LOS as proposed.

| SI 65 | Figure S4 shows the spatial allocation of valid counting stations. In total, 82 road links have valid counting data assigned to them. |
| --- | --- |
| 386 | Figure 10 shows the analysis result. A systematic, overall positive bias can be observed for the hourly, daily, and annual traffic volume (Fig. 10 b, d, and f). Counting stations on Distributor/Secondary roads tend to show higher values, while the model slightly overestimates the traffic volume on primary city roads and the motorway with high traffic volumes. However, it is also possible that the traffic counting stations, which are taken as the ground truth in this analysis, underestimate the volume of traffic, particularly at high volumes, e.g., due to incorrect or missing counts, or the malfunctioning of individual detectors.

[Figure]

 **Figure S4.** Classification and spatial allocation of "valid" and "invalid" flagged traffic counting stations. Valid stations provide data for 2019 and have an SQV greater than 0.6. In total, 82 road links have valid traffic counting data assigned to them. |

[Figure]

**Figure 10.** Uncertainty assessment of the activity data used in the present study: (a) and (b) show how hourly measured traffic counts match the modeled traffic volume for the total traffic at 82 road links classified by the road type. (c) and (d) illustrate the same for daily and (e,f) for annual traffic volumes. The model seems to overestimate the traffic volume at higher levels (Motorway and TrunkRoad/Primary-City) and underestimate the traffic at lower-level roads (Distributor/Secondary). The 95 % confidence interval of the relative uncertainty significantly decreases with temporal aggregation, as expected. Considering the large extent of the traffic model and the high number of counting stations, these numbers can be well accepted.

8. Reviewer comment: The comparison in Table 4 is useful but could be more diagnostic. Splitting differences by road class or simple urban/rural zones would help disentangle whether mismatches are driven by spatial allocation or by speed/EF assumptions.

**Response:** The UBA and TNO inventories are only available in a gridded form, which does not allow for disentangling differences at a road link level. To provide further diagnostics, we analysed which road type contributes the highest share of $CO_2$ emissions to each grid cell in our inventory. This results in five categories: "Motorway", "Primary", "Secondary", "Residential", and "None". The "None" category indicates cells where no emissions were allocated in our inventory. The analysis was added to the table.

335

**Table 7.** Comparison of the total emission for fossil fuel $CO_2$ ($CO_{2,ff}$), fossil and biofuel $CO_2$ ($CO_{2,ff+bf}$), $CO$ and $NO_x$ from three different spatially explicit emission inventories in Munich (DRIVE, UBA, TNO). For this comparison, we selected the closest year available. The RKU estimate is the official number reported by the City of Munich and includes upstream emissions from the fuel supply chain (Scope 2). We compare the $CO_{2e,WTW}$ (Well-to-Wheel) emission in this case. All other emissions are tank-to-wheel, i.e., Scope 1 emissions. In addition, the table contains total values for $CO_{2,ff+bf}$ for subsets categorized according to the predominant road types in the grid cell. This suggests that UBA overestimates emissions at Secondary roads and TNO underestimates emissions on the Motorway.

| Component | Unit | DRIVE (2019) | UBA (2019) | | TNO (2018) | | RKU (2019) | |
|---|---|---|---|---|---|---|---|---|
| $CO_{2,ff}$ | kt | 1248 | - | - | 946 | - 24% | | |
| $CO_{2,ff+bf}$ | kt | 1312 | 1936 | + 47% | 993 | - 24% | | |
| $CO$ | t | 4194 | 16250 | + 287% | 7671 | + 83% | | |
| $NO_x$ | t | 3434 | 5299 | + 54% | 2946 | - 14% | | |
| $CO_{2e,WTW}$ | kt | 1499 | - | - | - | - | 1592 | + 6.2 % |
| $CO_{2,ff+bf}$, Motorway | kt | 513 | 551 | + 7% | 235 | - 54% | | |
| $CO_{2,ff+bf}$, Primary | kt | 220 | 263 | + 20% | 170 | - 23% | | |
| $CO_{2,ff+bf}$, Secondary | kt | 489 | 974 | + 99% | 489 | 0% | | |
| $CO_{2,ff+bf}$, Residential | kt | 27 | 94 | + 248% | 25 | - 7% | | |
| $CO_{2,ff+bf}$, None | kt | 0 | 53 | - | 27 | - | | |
| $CO_{2,ff+bf}/NO_x$ | | 363.4 | 365.4 | | 321.1 | | - | |
| $CO_{2,ff+bf}/CO$ | | 297.6 | 119.1 | | 123.3 | | - | |

**Sources**

Forschungsgesellschaft für Straßen- und Verkehrswesen FSGV (Ed.). (2015). Handbuch für die Bemessung von Straßenverkehrsanlagen: HBS 2015. Teil S - Stadtstraßen (Ausg. 2015, Stand: 18.9.2015). FGSV-Verl.

Pina, N. and Tchepel, O.: A Bottom-up Modeling Approach to Quantify Cold Start Emissions from Urban Road Traffic, International Journal of Sustainable Transportation, 17, 942–955, https://doi.org/10.1080/15568318.2022.2130841, 2023.